# Source-Resolved Volatility and Oxidation State Decoupling in Wintertime Organic Aerosols in Seoul

Hwajin Kim[1,2,*] Jiwoo Jeong[1], Jihye Moon[1], Hyun Gu Kang[2,3]

[1]Department of Environmental Health Sciences, Graduate School of Public Health, Seoul National University, 08826 Seoul, South Korea
[2]Institute of Health and Environment, Graduate School of Public Health, Seoul National University, 08826 Seoul, South Korea
[3]Now at Multiphase Chemistry Department, Max Planck Institute for Chemistry, 55128 Mainz, Germany

*Correspondence to*: Hwajin Kim (khj0116@snu.ac.kr)

**Abstract.**

Organic aerosols (OA) are key components of wintertime urban haze, but the relationship between their oxidation state and volatility—critical for understanding aerosol evolution and improving model predictions—remains poorly constrained. While oxidation–volatility decoupling has been observed in laboratory studies, field-based evidence under real-world conditions is scarce, particularly during severe haze episodes. This study presents a field-based investigation of OA sources and their volatility characteristics in Seoul during a winter haze period, using a thermodenuder coupled with a high-resolution time-of-flight aerosol mass spectrometer (HR-ToF-AMS). Positive matrix factorization resolved six OA factors: hydrocarbon-like OA, cooking, biomass burning, nitrogen-containing OA (NOA), less-oxidized oxygenated OA (LO-OOA), and more-oxidized OOA (MO-OOA). Despite having the highest oxygen-to-carbon ratio (~1.15), MO-OOA exhibited unexpectedly high volatility, indicating a decoupling between oxidation state and volatility. We attribute this to fragmentation-driven aging and autoxidation under stagnant conditions with limited OH exposure. In contrast, LO-OOA showed lower volatility and more typical oxidative behavior.

Additionally, NOA—a rarely resolved factor in wintertime field studies—was prominent during cold, humid, and stagnant conditions and exhibited chemical and volatility features similar to biomass burning OA, suggesting a shared combustion origin and meteorological sensitivity.

These findings provide one of the few field-based demonstrations of oxidation–volatility decoupling in ambient OA and highlight how source-specific properties and meteorology influence OA evolution. The results underscore the need to refine OA representation in chemical transport models, especially under haze conditions.

Keywords: Organic aerosol volatility, HR-ToF-AMS, Thermodenuder, elemental ratios, aging, fragmentation

## 1 Introduction

Atmospheric aerosols affect both human health and the environment by reducing visibility (Ghim et al., 2005; Zhao et al., 2013) and contributing to cardiovascular and respiratory diseases (Hamanaka et al., 2018; Manisalidis et al., 2020). In addition, aerosols play a significant role in climate change by scattering or absorbing solar radiation and modifying cloud properties (IPCC AR6). Among the various aerosol components—including sulfate, nitrate, ammonium, chloride, crustal materials, and water—organic aerosols (OA) are particularly important to characterize, as they account for 20–90% of submicron particulate matter (Zhang et al., 2007). Identifying OA sources and understanding their behavior are critical for effective air quality management; however, this is particularly challenging due to the vast diversity and dynamic nature of OA compounds, which originate from both natural and anthropogenic sources. Unlike inorganic aerosols, organic aerosols (OAs) evolve continuously through complex atmospheric reactions, influenced by emission sources, meteorological conditions, and aerosol properties (Jimenez et al., 2009; Hallquist et al., 2009; Robinson et al., 2007; Donahue et al., 2006; Ng et al., 2010; Cappa and Jimenez, 2010).

Volatility is a key parameter for characterizing organic aerosol (OA) properties, as it governs gas-to-particle partitioning behavior and directly influences particle formation yields (Sinha et al., 2023). The classification of OA species based on their volatility—from extremely low-volatility (ELVOC) to semi-volatile (SVOC) and intermediate-volatility (IVOC) compounds—is central to the conceptual framework of secondary OA (SOA) formation and growth (Donahue et al., 2006). It also affects atmospheric lifetimes and human exposure by determining how long aerosols remain suspended in the atmosphere (Glasius and Goldstein, 2016). Therefore, accurately capturing OA volatility is essential for improving predictions of OA concentrations and their environmental and health impacts. However, chemical transport models often significantly underestimate OA mass compared to observations (Jiang et al., 2012; Li et al., 2017), largely due to incomplete precursor inventories and simplified treatment of processes affecting OA volatility. For instance, aging—through oxidation reactions such as functionalization and fragmentation—can significantly alter volatility by changing OA chemical structure (Robinson et al., 2007; Zhao et al., 2016). Early volatility studies primarily utilized thermal denuders (TD) coupled with various detection instruments to investigate the thermal properties of bulk OA (Huffman et al., 2008). The subsequent coupling of TD with the Aerosol Mass Spectrometer allowed for component-resolved volatility measurements, providing critical, quantitative insight into the properties of OA factors (e.g., SV-OOA vs. LV-OOA) across different regions (Paciga et al., 2016; Cappa and Jimenez, 2010). These component-resolved volatility data are often used to constrain the Volatility Basis Set (VBS)—the current state-of-the-art framework for

modeling OA partitioning and evolution (Donahue et al., 2006). However, a limitation in many field studies is that the TD-AMS thermogram data are rarely translated into quantitative VBS distributions for individual OA factors, which limits their direct use in chemical transport models. Furthermore, the volatility of OOA during extreme haze conditions, where the expected inverse correlation between oxidation (O:C) and volatility can break down (Jimenez et al., 2009), remains poorly characterized, particularly in East Asia's highly polluted winter environments. A recent study in Korea further highlighted the importance of accounting for such processes when interpreting OA volatility under ambient conditions (Kang et al., 2022). Given its central role in OA formation, reaction, and atmospheric persistence, volatility analysis is critical for bridging the gap between measurements and model performance.

Traditionally, due to the complexity and variability of OA, the oxygen-to-carbon (O:C) ratio has been used as a proxy for estimating volatility. In general, higher O:C values indicate greater oxidation and lower volatility (Jimenez et al., 2009). Accordingly, many field studies classify oxygenated OA (OOA) into semi-volatile OOA (SV-OOA) and low-volatility OOA (LV-OOA) based on their O:C ratios (Ng et al., 2010; Huang et al., 2010; Mohr et al., 2012). However, this relationship is not always straightforward. Fragmentation during oxidation can increase both O:C and volatility simultaneously, disrupting the expected inverse correlation (Jimenez et al., 2009). In laboratory experiments, yields of highly oxidized SOA have been observed to decrease due to fragmentation (Xu et al., 2014; Grieshop et al., 2009). These findings suggest that while O:C can offer useful insights, it is insufficient alone to represent OA volatility. Direct volatility measurements, especially when paired with chemical composition data, are necessary to improve our understanding of OA sources and aging processes.

In this study, we investigate the sources and volatility characteristics of OA in Seoul during winter. Wintertime OA presents additional challenges due to its high complexity. During winter, emissions from combustion sources such as biomass burning and residential heating significantly increase, contributing large amounts of primary OA (Kim et al., 2017). Meanwhile, low ambient temperatures and reduced photochemical activity affect the formation and evolution of secondary OA (SOA). Frequent haze events further complicate the aerosol properties by extending aging times and increasing particle loadings. These overlapping sources and atmospheric conditions make winter OA particularly difficult to characterize and predict. Despite Seoul's significance for air quality management, comprehensive studies on OA volatility during winter remain limited. To address these goals, we conducted real-time, high-resolution measurements using a high-resolution time-of-flight aerosol mass spectrometer (HR-ToF-AMS) coupled with a thermodenuder (TD). The objectives of this study are to: (1) improve the understanding of

wintertime OA in Seoul, (2) characterize the volatility of OA associated with different sources, and (3) explore the
relationship between OA volatility and chemical composition.

## 2 Experimental methods

### 2.1 Sampling Site and Measurement Period

We conducted continuous real-time measurements in Seoul, South Korea, from 28 November to 28 December
2019. The sampling site was located in the northeastern part of the city (37.60° N, 127.05° E), approximately 7 km
from the city center, surrounded by major roadways and mixed commercial–residential land use. Air samples were
collected at an elevation of approximately 60 meters above sea level, on the fifth floor of a building. A detailed site
description has been reported previously for winter Seoul (Kim et al., 2017). During this period, the average
ambient temperature was $1.76 \pm 4.3$ °C, and the average relative humidity (RH) was $56.9 \pm 17.5\%$, based on data
from the Korea Meteorological Administration (http://www.kma.go.kr).

### 2.2 Instrumentation and Measurements

The physico-chemical properties of non-refractory $PM_1$ (NR-$PM_1$) species—including sulfate, nitrate, ammonium,
chloride, and organics—were measured using an Aerodyne high-resolution time-of-flight aerosol mass
spectrometer (HR-ToF-AMS) (DeCarlo et al., 2006). $PM_1$ mass in this study is taken as NR-$PM_1$ (from AMS) +
black carbon (BC; measured by MAAP), which is appropriate for winter Seoul where refractory $PM_1$ (metal/sea-
salt/crustal) is minor and dust events were excluded (e.g., Kim et al., 2017; Nault et al., 2018; Kang et al., 2022;
Jeon et al., 2023).Data were acquired at 2.5-minute intervals, alternating between V and W modes. The V mode
provides higher sensitivity but lower resolution, suitable for mass quantification, whereas the W mode offers higher
mass resolution but lower sensitivity, used here for OA source apportionment. Simultaneously, black carbon (BC)
concentrations were measured at 1-minute intervals using a multi-angle absorption photometer (MAAP; Thermo
Fisher Scientific, Waltham, MA, USA). Total $PM_1$ mass was calculated as the sum of NR-$PM_1$ and BC.
Hourly trace gas concentrations (CO, $O_3$, $NO_2$, $SO_2$) were obtained from the Gireum air quality monitoring station
(37.61° N, 127.03° E), managed by the Seoul Research Institute of Public Health and Environment. Meteorological
data (temperature, RH, wind speed/direction) were collected from the nearby Jungreung site (37.61° N, 127.00°
E). All data are reported in Korea Standard Time (UTC+9).
To examine aerosol volatility, a thermodenuder (TD; Envalytix LLC) was installed upstream of the HR-ToF-AMS.
Details are provided in Supplementary Section S1 Kang et al. (2022). Briefly, ambient flow alternated every 5
minutes between a TD line and a bypass line at 1.1 L min$^{-1}$. Residence time in the TD line was ~6.3 s. The TD
setup included a 50 cm heating section followed by an adsorption unit. Heated particles were stripped of volatile
species, while the downstream carbon-packed section prevented recondensation. TD temperature cycled through
12 steps (30 to 200 °C), with each step lasting 10 min (total cycle = 120 min). AMS V and W modes were alternated
during the same cycle. The heater was pre-adjusted to the next temperature while the bypass was active.

**2.3 Data Analysis**

**2.3.1 Data analysis and OA Source Apportionment**
HR-AMS data were processed using SQUIRREL v1.65B and PIKA v1.25B. Mass concentrations of non-refractory
PM$_1$ (NR-PM$_1$) species were derived from V-mode data, while high-resolution mass spectra (HRMS) and the
elemental composition of organic aerosols (OA) were obtained from W-mode data. NR-PM$_1$ quantification
followed established AMS protocols (Ulbrich et al., 2009; Zhang et al., 2011). Both the bypass and TD streams
were processed using a time-resolved, composition-dependent collection efficiency CE(t) following Middlebrook
et al. (2012). TD heating can modify particle water and phase state/mixing and thereby influence CE beyond
composition (Huffman et al., 2009), but prior TD–AMS studies indicate that such effects are modest and largely
multiplicative, which do not distort thermogram shapes or T$_{50}$ ordering (Faulhaber et al., 2009; Cappa & Jimenez,
2010). In our data, the CE(t) statistics for the two lines were similar (campaign-average CE: TD = 0.55 ± 0.08;
bypass = 0.53 ± 0.04; Δ = 0.02 ≈ 3.7%, below the combined uncertainty ≈ 0.09). We therefore report volatility
metrics with these line-specific CE(t) corrections applied and interpret potential residual CE effects as minor. For
organics,elemental ratios (O:C, H:C, and OM/OC) were calculated using the Improved-Ambient (IA) method
(Canagaratna et al., 2015). Positive Matrix Factorization (PMF) was applied to the HRMS of organics using the
PMF2 algorithm (v4.2, robust mode) (Paatero and Tapper, 1994). The HRMS and corresponding error matrices
from PIKA were analyzed using the PMF Evaluation Tool v2.05 (Ulbrich et al., 2009). Data pretreatment followed
established protocols (Ulbrich et al., 2009; Zhang et al., 2011). A six-factor solution (fPeak = 0; Q/Q_expected =
3.56) was selected as optimal (Fig. S1). The resolved OA sources included hydrocarbon-like OA (HOA; 14%; O:C
= 0.13), cooking-related OA (COA; 21%; O:C = 0.18), nitrogen-enriched OA (NOA; 2%; O:C = 0.22), biomass-
burning OA (BBOA; 13%; O:C = 0.25), less-oxidized oxygenated OA (LO-OOA; 30%; O:C = 0.68), and more-
oxidized oxygenated OA (MO-OOA; 20%; O:C = 1.15) (Figs. S2 and S3). Alternative five- and seven-factor
solutions were also evaluated. In the five-factor solution, the biomass burning source was not clearly resolved and
appeared to be distributed across multiple factors. In the seven-factor solution, BBOA was further split into two
separate factors without clear distinction or added interpretive value, making the six-factor solution the most
physically meaningful and interpretable (Figs. S4 and S5). To ensure the statistical robustness of this solution, we
calculated uncertainties for each PMF factor using the bootstrap method (100 iterations) with the PET toolkit
(v2.05) (EPA, 2014; Xu et al., 2018; Srivastava et al., 2021) (Table S2 and Fig. S13).

## 2.3.2 Thermogram and Volatility Estimation

The chemical composition dependent mass fraction remaining (MFR) was derived at each TD temperature by
dividing the corrected mass concentration of the TD line [p] by the average of the adjacent bypass lines [p-1] and
[p+1]. Thermograms were corrected for particle loss, estimated using reference substances like NaCl, which exhibit
minimal evaporation (Huffman et al., 2009; Saha et al., 2014; Kang et al., 2022). OA factor concentrations at each
TD temperature were derived via multivariate linear regression between post-TD HRMS and ambient OA factor
HRMS profiles as described in Zhou et al., 2017.
Volatility distributions were modeled using the thermodenuder mass transfer model from Riipinen et al. (2010) and
Karnezi et al. (2014), implemented in Igor Pro 9 (Kang et al., 2022). OA mass was distributed into eight logarithmic
saturation concentration bins (C*: 1000 to 0.0001 $\mu g\,m^{-3}$). Modeled MFRs were fit to observations using Igor's
"FuncFit" function, repeated 1,000 times per OA factor to determine best-fit results. The model assumes no thermal
decomposition and includes adjustable parameters: mass accommodation coefficient ($\alpha_m$) and enthalpy of
vaporization ($\Delta H_{exp}$), randomly sampled within literature-based ranges (Table S1).

## 3 Results and discussion

## 3.1 Overview of PM$_1$ Composition and OA Sources

We conducted continuous measurements from 28 November to 28 December 2019, characterizing a winter period
with a mean PM$_1$ concentration of $27.8\pm15.3\,\mu g\,m^{-3}$. This concentration is characterized as moderate; it closely
matches historical winter PM$_1$ means in Seoul (Kim et al., 2017) and implies an equivalent PM$_{2.5}$ concentration is
about 34.8μgm$^{-3}$ (using a Korea-specific PM$_1$/PM$_{2.5}$≈0.8 (Kwon et al., 2023), which is near the national 24-h PM$_{2.5}$
standard (35μgm$^{-3}$) (AirKorea). The full co-evolution of PM$_1$, gaseous pollutants, and meteorological conditions
is provided in Fig. S6, showing an average ambient temperature of 1.76±4.3∘C and average relative humidity (RH)
of 56.9±17.5% during the study.
Figure 1 summarizes the overall non-refractory submicron aerosol (NR-PM$_1$) composition and the identified OA
factors. Organics (41%) and nitrate (30%) were the most abundant chemical components of PM$_1$, followed by
ammonium (12%), sulfate (10%), BC (5%), and chloride (3%) (Fig. 1a). Among the organic aerosols, six OA
factors were identified during the winter of 2019: hydrocarbon-like OA (HOA; 14%; O:C = 0.13), cooking-related
OA (COA; 21%; O:C = 0.18), nitrogen-enriched OA (NOA; 2%; O:C = 0.22), biomass burning OA (BBOA; 13%;
O:C = 0.25), and two types of secondary organic aerosols—less-oxidized oxygenated OA (LO-OOA; 30%; O:C =
0.68) and more-oxidized oxygenated OA (MO-OOA; 20%; O:C = 1.15) (Fig. 1e and Fig. S2). These compositions
are consistent with previous wintertime observations in Kim et al. (2017), with the exception of newly resolved
NOA source. In the following sections, we describe each OA factor in the order of secondary OA (SOA), primary
OA (POA) and finally introduce NOA, which—while related to combustion POA—emerged as a distinct, nitrogen-
rich factor under the winter conditions of this study.
PM$_1$ mass concentrations varied widely, ranging from 4.61 to 91.4 μg m$^{-3}$, largely due to two severe haze episodes
that occurred between December 7–12 and December 22–26 (Fig. 1). During these episodes, average
concentrations increased significantly, driven primarily by elevated levels of nitrate and organic aerosols—
particularly MO-OOA and NOA (Fig. 1f,g). Back-trajectory clustering shows frequent short-range recirculation
over the Seoul Metropolitan Area during haze (Cluster 1; Fig. S8), and the time series indicates persistently low
surface wind speeds during these periods (1.73 ± 0.89 vs. 2.34 ± 1.18 (clean)) (Fig. S6). These patterns indicate
stagnation-driven accumulation of local emissions, consistent with the simultaneous increase of MO-OOA and
NOA that are examined in detail in subsequent sections. Such haze episodes, characterized by local emission
buildup and secondary aerosol production, are a typical wintertime feature, as also reported in Kim et al. (2017).
**3.1.1 Secondary organic aerosols (SOA)**
In this study, two OOA factors—more-oxidized OOA (MO-OOA) and less-oxidized OOA (LO-OOA)—were
identified, together accounting for approximately half of the total organic aerosol (OA) mass. This fraction is
notably higher than that reported in previous wintertime urban studies (Kim et al., 2017; Zhang et al., 2007). Both
OOAs exhibited characteristic mass spectral features, including prominent peaks at $m/z$ 44 ($CO_2^+$) and $m/z$ 43
($C_2H_3O^+$), which are widely recognized as markers of oxygenated organics (Fig. S2e, S3f). The oxygen-to-carbon
(O:C) ratios for MO-OOA and LO-OOA were 1.15 and 0.68, respectively, indicating both factors are highly
oxidized relative to the primary OA factors (HOA, COA, BBOA) and that MO-OOA is substantially more oxidized
than LO-OOA. The O:C ratio of MO-OOA was especially elevated, exceeding those reported in previous Seoul
campaigns—0.68 in winter 2015 (Kim et al., 2017), 0.99 in spring 2019 (Kim et al., 2020), and 0.78 in fall 2019
(Jeon et al., 2023)—while the LO-OOA ratio was within a similar range.
MO-OOA showed strong correlations with secondary inorganic species such as nitrate (r = 0.90), ammonium (r =
0.92), and sulfate (r = 0.81), consistent with its formation through regional and local photochemical aging processes
(Fig. S3). In contrast, LO-OOA exhibited only modest correlations with sulfate, nitrate, and ammonium ($r = 0.50$,
0.51, and 0.42, respectively). This weaker coupling indicates that LO-OOA represents a less aged oxygenated OA
component (fresh SOA), distinguishable from the more aged, highly processed MO-OOA which tracks closely with
secondary inorganic species. Regarding potential primary influence, LO-OOA does not exhibit a pronounced $m/z$
60 (levoglucosan) signal (Figs. S2 and 9). While the levoglucosan marker ($f_{60}$) is known to diminish with
atmospheric aging and can become weak or undetectable downwind (Hennigan et al., 2010; Cubison et al., 2011),
the absence of a distinct peak combined with the separation from inorganic salts suggests that LO-OOA is best
characterized as freshly formed secondary organic aerosol likely originating from the rapid oxidation of local
anthropogenic precursors.

### 217 3.1.2 Primary organic aerosols (POA)

Three primary organic aerosol (POA) factors were identified in this study: hydrocarbon-like OA (HOA), cooking-
related OA (COA), and biomass burning OA (BBOA). These three components exhibited mass spectral and
temporal characteristics consistent with previous observations in Seoul and other urban environments. HOA was
characterized by dominant alkyl fragment ions ($C_nH_{2n+1}^+$ and $C_nH_{2n-1}^+$; Fig. S2a) and a low O:C ratio (0.13),
consistent with traffic-related emissions (0.05–0.25) (Canagaratna et al., 2015). It showed strong correlations with
vehicle-related ions $C_3H_7^+$ (r = 0.79) and $C_4H_9^+$ (r = 0.86) (Kim et al., 2017; Canagaratna et al., 2004; Zhang et al.,
2005), and exhibited a distinct morning rush hour peak (06:00–08:00), followed by a decrease likely driven by
boundary layer expansion (Fig. S3a).
COA, accounting for 21% of OA, showed higher contributions from oxygenated ions than HOA, with tracer peaks
at $m/z$ 55, 84 and 98 (Fig. S2b) consistent with cooking emissions (Sun et al., 2011). COA showed an enhanced
signal at $m/z$ 55 relative to $m/z$ 57, with a 55/57 ratio of 3.11, substantially larger than that of HOA (1.10). This
elevated ratio is consistent with previously reported AMS COA spectra in urban environments (e.g., Allan et al.,
2010; Mohr et al., 2012; Sun et al., 2011), supporting our factor assignment. It correlated strongly with cooking-
related ions such as $C_3H_3O^+$ (r = 0.94), $C_5H_8O^+$ (r = 0.96), and $C_6H_{10}O^+$ (r = 0.98) (Fig. S3h), and displayed
prominent peaks during lunch and dinner hours, reflecting typical cooking activity patterns.
BBOA was identified based on characteristic ions at $m/z$ 60 ($C_2H_4O_2^+$) and 73 ($C_3H_5O^+$), both of which are
associated with levoglucosan—a well-established tracer for biomass burning (Simoneit et al., 2002). Its relatively
high $f_{60}$ and low $f_{44}$ values (Fig. S9) indicate that the BBOA observed in this study was relatively fresh and had not
undergone extensive atmospheric aging (Cubison et al., 2011). Regarding source location, several pathways can
influence Seoul's biomass burning signature. First, urban/peri-urban small-scale burning (e.g., solid-fuel use in
select households, restaurant charcoal use, and intermittent waste burning) has been reported and can enhance
BBOA locally (Kim et al., 2017). Second, nearby agricultural-residue burning in surrounding provinces occurs
seasonally and can episodically impact the metropolitan area (Han et al., 2022). Third, regional transport from
upwind regions (e.g., northeastern China/North Korea) can bring biomass burning influenced air masses under
northerly/northwesterly flow (Lamb et al., 2018; Nault et al., 2018). In this dataset, the nighttime and early-
morning enhancements and trajectory clusters showing regional recirculation indicate a predominantly local/near-
source contribution during the study period, with episodic non-local influences remaining possible (Fig. S8).

### 3.1.2.1 Nitrogen-containing organic aerosol (NOA)

A distinct nitrogen-containing organic aerosol (NOA) factor was resolved in this study, whereas earlier wintertime
AMS–PMF analyses in Seoul did not isolate such a component. The NOA factor exhibited the highest nitrogen-to-
carbon (N:C) ratio (0.22) and the lowest oxygen-to-carbon (O:C) ratio (0.19) among all POA factors (Fig. S2),
indicating a chemically reduced, nitrogen-rich composition. The NOA mass spectrum was dominated by amine-
related fragments including $m/z$ 30 ($CH_4N^+$), 44 ($C_2H_6N^+$), 58 ($C_3H_8N^+$), and 86 ($C_5H_{12}N^+$) (Fig. 3a). The spectral
signature of the factor is defined by the characteristic dominance of the $m/z$ 44 fragment, which typically serves as

the primary marker for dimethylamine (DMA)-related species, closely followed by $m/z$ 58 (trimethylamine, TMA) and $m/z$ 30 (methylamine, MA). This profile is in strong agreement with NOA factors resolved via PMF in other polluted environments. For instance, the dominance of $m/z$ 44 and $m/z$ 30 aligns with amine factors reported in New York City (Sun et al., 2011) and Pasadena, California (Hayes et al., 2013). This DMA-dominated signature is also consistent with seasonal characterization of organic nitrogen in Beijing (Xu et al., 2017) and Po Valley, Italy (Saarikoski et al., 2012), reinforcing the common chemical signature of reduced organic nitrogen across diverse urban and regional environments.

In this study, NOA contributed approximately 2 % of total OA, comparable to urban contributions reported in Guangzhou (3 %; Chen et al., 2021), Pasadena (5 %; Hayes et al., 2013), and New York (5.8 %; Sun et al., 2011). These similarities suggest that the NOA factor observed in Seoul reflects a broader class of urban wintertime reduced-nitrogen aerosols rather than a site-specific anomaly. Furthermore, the presence of non-negligible signals at m/z 58 and m/z 86 supports the contribution of slightly larger alkylamines, a pattern that aligns well with established AMS laboratory reference spectra (Ge et al., 2011; Silva et al., 2008). In most urban environments, the detectability of NOA appears to depend strongly on the interplay between emission strength, stagnation, and humidity—which together govern the particle-phase partitioning of volatile amines.

These amines are commonly emitted during the combustion of nitrogen-rich biomass and proteinaceous materials and are frequently associated with biomass-burning emissions (Ge et al., 2011). Previous molecular analyses in Seoul also indicate DMA, MA, and TMA as the dominant amine species in December (Baek et al., 2022). While other amines such as triethylamine (TEA), diethylamine (DEA), and ethylamine (EA) may contribute via industrial/solvent pathways (e.g., chemical manufacturing, petrochemical corridors, wastewater treatment), our HR-AMS spectra are dominated by small alkylamine fragments ($m/z$ 30, 44, 58, 86) and the diurnal behavior co-varies with combustion markers (Fig. 2), indicating a primarily combustion-linked influence. Nevertheless, recent urban measurements and sector-based analyses show that industrial activities can contribute measurable amines in cities (Tiszenkel et al., 2024; Zheng et al., 2015; Mao et al., 2018; Shen et al., 2017; Yao et al., 2016). Accordingly, a minor NOA contribution from solvent/industrial amines cannot be excluded. NOA exhibited a nighttime–early-morning enhancement (Fig. 2a), similar to BBOA, indicating that both factors are influenced by wintertime combustion and residential heating, which are known sources of small alkylamines and amides (You et al., 2014; Yao et al., 2016). Strong correlations of NOA with $CH_4N^+$ (r = 0.95) and $C_2H_6N^+$ (r = 0.91) (Fig. 2) further support the presence of reduced-nitrogen species associated with these combustion activities. However, the time series of

NOA and BBOA are not strongly correlated (Fig. 2 and Fig. S7). This contrast reflects their differing behaviors:
BBOA follows a relatively regular daily emission pattern, whereas NOA appears predominantly during stagnant
haze periods (Fig. 1) when cold, humid, and low-wind conditions allow semi-volatile amines to partition to the
particle phase and form low-volatility aminium salts. Thus, NOA in wintertime Seoul likely reflects a combination
of shared primary combustion influences and enhanced secondary processing of amine-containing precursors under
meteorological conditions that favor partitioning and accumulation.
Detection of particulate NOA using real time measurement has been challenging due to its low concentration and
high volatility. Although Baek et al. (2022) identified nitrogen-containing species in Seoul via year-round filter-
based molecular analysis, PMF-based resolution of NOA in real time has not been previously reported. The
successful identification in this study is likely attributable to favorable winter meteorological conditions—
specifically low temperatures (–0.24 °C) and persistently high relative humidity (~57%) compared to the 2017
winter season (Kim et al., 2017)—that enhanced gas-to-particle partitioning of semi-volatile amines, thereby
enabling their detection (Fig. S2). NOA concentrations frequently exceeded 1 $\mu g\ m^{-3}$ when RH surpassed 60% (Fig.
2), supporting the importance of RH-driven partitioning and the subsequent formation of low-volatility aminium
salts (Rovelli et al., 2017). Although extremely low temperatures may inhibit NOA formation due to the transition
of aerosol particles into solid phase (Ge et al., 2011; Srivastava et al., 2022), the combination of consistently cold
and humid conditions during the measurement period likely promoted the partitioning of semi-volatile amines into
the particle phase. In addition, episodic haze events further elevated NOA levels, increasing its contribution to OA
from 1% during clean periods to as much as 3% (Fig. 1f–h). These high-concentration events likely improved the
signal-to-noise ratio, facilitating PMF resolution. Back-trajectory clustering indicates that NOA-enhanced events
were dominated by short-range recirculation (Cluster 1; Fig. S7), consistent with the short atmospheric lifetimes
and high reactivity of alkylamines (Nielsen et al., 2012; Hanson et al., 2014). Overall, the factor reflects semi-
volatile, reduced-nitrogen species originating from primary urban combustion sources, with their observed particle-
phase mass amplified by rapid secondary partitioning and salt formation under seasonally favorable conditions.

## 3.2 Volatility of Non-Refractory Species

Figure 4 presents thermograms of non-refractory (NR) species measured by HR-ToF-AMS. The mass fraction
remaining (MFR) after thermodenuder (TD) treatment follows the typical volatility trend reported in previous
studies (Xu et al., 2016; Kang et al., 2022; Jeon et al., 2023; Huffman et al., 2009): nitrate was the most volatile,

followed by chloride, ammonium, organics, and sulfate. Nitrate showed the steepest decline with increasing temperature, with a $T_{50}$ of ~67 °C—substantially higher than that of pure ammonium nitrate (~37 °C; Huffman et al., 2009). At 200 °C, ~2% of the initial nitrate signal remained (Fig. 4). Since pure ammonium nitrate fully evaporates well below this temperature (Huffman et al., 2009), this small residual fraction likely represents the least volatile portion of organic nitrates. Compared to previously reported fall conditions ($T_{50}$ ~73 °C, incomplete evaporation), winter nitrate appeared more volatile, indicating relatively fewer non-volatile nitrate forms (e.g., Kang et al., 2022; Jeon et al., 2023). Sulfate exhibited the highest thermal stability among the measured species. The thermogram showed a relatively stable mass fraction (MFR > 0.8) up to ~130 °C, followed by a sharp decline at temperatures above 140 °C (Fig. 4). This profile is consistent with the typical volatilization behavior of ammonium sulfate in TD-AMS, which requires higher temperatures to evaporate compared to nitrate or organics (Huffman et al., 2009). At 200 °C, approximately 25% of the sulfate mass remained. This residual suggests the presence of a sulfate fraction with lower volatility than pure ammonium sulfate, likely associated with organosulfates or low-volatility mixtures, whereas refractory metal sulfates are not efficiently detected by the AMS (Canagaratna et al., 2007). Ammonium showed intermediate volatility, with $T_{50}$ between nitrate and sulfate. Its slightly lower winter $T_{50}$ suggests stronger nitrate association. Residual ammonium at 200 °C was consistent (~4%) in previously reported spring/fall measurements (Kang et al., 2022; Jeon et al., 2023). Chloride volatility was broadly consistent with prior AMS studies, with $T_{50}$ values comparable across seasons (e.g., Xu et al., 2016; Jeon et al., 2023). The near-complete evaporation observed in winter (~4% residual at 200 °C, Fig. 4) indicates that the chloride measured here was dominated by volatile inorganic chloride, specifically ammonium chloride ($NH_4Cl$), which fully evaporates at relatively low temperatures (Huffman et al., 2009). By contrast, metal chlorides (e.g., NaCl, KCl) are refractory and far less volatile; they are also poorly detected by the AMS (Canagaratna et al., 2007). The lower residual in winter compared to fall (~10%) therefore suggests that wintertime chloride consisted almost exclusively of pure ammonium chloride, whereas the fall samples may have contained a minor fraction of less volatile or refractory chloride species. Organics exhibited moderate volatility ($T_{50}$ ~120 °C), and their thermogram showed a gradual, continuous decrease in mass fraction with increasing TD temperature. This smooth profile reflects the presence of a broad distribution of organic compounds spanning SVOC to LVOC ranges, in contrast to inorganic species such as nitrate or ammonium chloride, which often show more abrupt losses at characteristic temperatures (Huffman et al., 2009; Xu et al., 2016). This behavior is consistent with previous TD-AMS observations in Seoul during spring and fall (Kang et al., 2022; Jeon et al., 2023).

### 3.2.1 Volatility Profiles of Organic sources

Figure 5 presents the volatility distributions of six OA sources within the volatility basis set (VBS) framework. Volatility is expressed as the effective saturation concentration ($C^*$, µg m$^{-3}$), where higher $C^*$ values correspond to higher volatility. Following Donahue et al. (2009), $C^*$ values are categorized into four bins: extremely low-volatility organic compounds (ELVOCs, log $C^* < -4.5$), low-volatility organic compounds (LVOCs, $-4.5 < $ log $C^* < -0.5$), semi-volatile organic compounds (SVOCs, $-0.5 < $ log $C^* < 2.5$), and intermediate-volatility organic compounds (IVOCs, $2.5 < $ log $C^* < 6.5$).

Among the primary OA (POA) sources, hydrocarbon-like OA (HOA) exhibited the highest volatility, with mass predominantly distributed in the SVOC and IVOC ranges, consistent with its chemically reduced nature (O:C = 0.13) and direct combustion origin. Mass fraction remaining (MFR) results (Fig. S9) further support this, showing rapid mass loss at lower temperatures. Biomass burning OA (BBOA) and nitrogen-containing OA (NOA) also showed high volatility, peaking in the SVOC–IVOC range (log $C^* = 1$–3), but displayed slightly higher O:C ratios (0.25 and 0.19, respectively). This modest enhancement in O:C reflects their source composition—biomass combustion produces partially oxygenated organic species (e.g., levoglucosan, phenols), and NOA contains nitrogen-bearing functional groups—rather than enhanced atmospheric oxidation. Cooking-related OA (COA) showed a more moderate volatility profile, with mass more evenly distributed across the LVOC and SVOC bins. This behavior differs from that of BBOA, which is slightly more oxidized yet more volatile. This apparent decoupling between oxidation state and volatility is a characteristic feature of COA reported in previous volatility studies (Paciga et al., 2016; Kang et al., 2022). These studies attribute the lower volatility of COA to its abundance of high-molecular-weight fatty acids (e.g., oleic, palmitic, and stearic acids) and glycerides (Mohr et al., 2009; He et al., 2010). Unlike the smaller, fragmented molecules typical of biomass burning, these lipid-like compounds possess high molar masses that suppress volatility, even though their long alkyl chains result in low O:C ratios.

For secondary OA (SOA), less-oxidized oxygenated OA (LO-OOA) exhibited the lowest volatility, with substantial mass in the LVOC and ELVOC bins ($C^* \approx 10^{-3}$–$10^{-4}$). This is in agreement with previous findings in Seoul during spring (Kang et al., 2022). In contrast, more-oxidized OOA (MO-OOA), despite its higher oxidation state (O:C =

1.15), displayed greater volatility, with a peak at $C^* \approx 10^1$. This discrepancy likely reflects differences in formation
and aging processes, as discussed further in Section 3.3.
Overall, the volatility characteristics across OA factors suggest that oxidation state alone does not fully explain
volatility. Rather, volatility is shaped by a combination of emission source, emission timing, temperature, and
atmospheric processing. These findings highlight the importance of integrating both chemical and physical
characterization to better understand OA formation and aging across seasons.

## 370  3.3 Aging effect on volatility from 2D VBS

Generally, the oxygen-to-carbon (O:C) ratio of organic aerosols (OA) is inversely related to their volatility. As O:C
increases through aging, the effective saturation concentration ($C^*$) typically decreases, resulting in lower volatility
(Donahue et al., 2006; Jimenez et al., 2009). This relationship arises because oxidative functionalization introduces
polar groups (e.g., hydroxyl, carboxyl) that increase molecular weight and enhance intermolecular hydrogen
bonding, thereby reducing the effective saturation concentration ($C^*$) and promoting particle-phase retention
(Jimenez et al., 2009; Kroll and Seinfeld, 2008; Donahue et al., 2011). However, in this study, the most oxidized
OA factor—MO-OOA, with a high O:C ratio of 1.15—exhibited unexpectedly high volatility. Its volatility
distribution was skewed toward SVOCs and IVOCs (Fig. 5), and its rapid mass loss in MFR thermograms (Fig.
S9) further indicated low thermal stability. This observation appears to contradict the usual inverse O:C–volatility
relationship; however, under winter haze conditions—with suppressed $O_3$/low OH, particle-phase autoxidation and
fragmentation can yield higher-O:C yet more volatile products, with enhanced condensation on abundant particle
surface area (details below).
Viewed against prior TD-AMS results, the volatility of Seoul's winter MO-OOA presents a unique case,
particularly in the nature of its O:C-volatility relationship. Prior urban studies have commonly reported substantial
SVOC-OA, consistent with high photochemical activity or elevated loadings; for example, prior TD-AMS studies
in Mexico City, Los Angeles, Beijing, and Shenzhen have all reported substantial SVOC–IVOC contributions
during polluted periods, indicating that high OA volatility is a common feature of urban environments across
seasons (Cappa and Jimenez, 2010; Xu et al., 2019; Cao et al., 2018). While these comparisons establish that
volatile OA is common, they generally did not report the factor-level inversion observed here, where the highly-
oxidized OOA component (MO-OOA) was more volatile than a less-oxidized OOA (LO-OOA). This behavior is
distinct from findings in colder, lower-loading regimes; wintertime Paris, for instance, maintained the conventional
hierarchy where the more-oxidized OOA was comparatively less volatile (Paciga et al., 2016). Furthermore,
seasonal context within Seoul showed springtime OA with lower oxidation levels than our winter MO-OOA despite
similar SVOC contributions (Kang et al., 2022). This comprehensive comparison underscores the unusual nature
of the O:C-volatility relationship observed under the specific winter haze conditions in Seoul.
**3.3.1 High-volatility nature of MO-OOA in Seoul wintertime**
MO-OOA exhibited high O:C ratios and high apparent volatility, characteristics that were further amplified during
haze episodes—periods marked by reduced ozone levels, low solar radiation, and elevated aerosol mass
concentrations (Fig. 7 and Fig. S6, yellow shading). Spectrally, MO-OOA was defined by a consistently high $f_{44}$
($CO_2^+$) signal and a comparatively stable $f_{43}$ ($C_2H_3O^+$) signal relative to LO-OOA (Fig. 6). Notably, when MO-
OOA concentrations intensified during haze, only $f_{44}$ was significantly enhanced, while $f_{43}$ remained nearly
unchanged (Fig. 6). This trend is corroborated by the haze–non-haze comparison (Fig. S12), where haze periods
(including high MO-OOA intervals) showed elevated contributions from oxygenated fragments ($m/z$ 28, 29, 44)
and higher O:C ratios. In contrast, non-haze periods were characterized by larger fractional contributions from
hydrocarbon-like fragments ($m/z$ 41, 43, 55, 57). The observed temporal pattern—elevated $f_{44}$ without
corresponding changes in $f_{43}$—is a typical signature of highly oxidized and fragmented organic aerosol (Figs. 6 and
7), suggesting that aging was dominated by fragmentation rather than functionalization (Kroll et al., 2009). These
spectral patterns collectively indicate that MO-OOA is highly oxidized yet remains relatively volatile compared to
LO-OOA.
The elevated volatility of MO-OOA despite its high O:C (~1.15) indicates that oxidation under these haze
conditions did not follow the classical multi-generational OH-driven aging pathway, which typically increases
molecular mass and reduces volatility. Instead, the data align with fragmentation-dominated aging, where highly
oxygenated but lower-molecular-weight compounds (e.g., small acids or diacids) are formed. Prior field and
laboratory studies using online AMS/FIGAERO-CIMS and EESI-TOF have similarly reported high-O:C yet
volatile product distributions characterized by high $f_{44}$ and stable $f_{43}$ (Kroll et al., 2009; Ng et al., 2010; Chhabra et
al., 2011; Lambe et al., 2012; Lopez-Hilfiker et al., 2016; D'Ambro et al., 2017).
While direct mechanistic measurements were not available in this study, we hypothesize that the formation of this
volatile, high-O:C component may be driven by specific low-light oxidation pathways consistent with the observed
environmental conditions. The suppressed ozone levels during haze likely indicate a low-OH oxidation regime
(Fig. 7). Under such conditions, radical chemistry involving $NO_3$ (which is longer-lived in low light) or particle-
phase autoxidation could preferentially produce highly oxygenated but relatively small organic fragments (Ehn et
al., 2014; Zhao et al., 2023). Although haze suppresses photolysis, HONO concentrations—maintained via
heterogeneous conversion or surface emissions—could still provide a non-negligible source of OH (Gil et al., 2021;
Kim et al., 2024; Slater et al., 2020). Furthermore, the high aerosol mass loadings during haze ($C_{oa}$) provide
abundant surface area for absorptive partitioning (Pankow, 1994; Donahue et al., 2006). This increased partitioning
mass allows even relatively volatile, oxidized compounds to condense into the particle phase, contributing to the
high apparent volatility and oxidation state observed (Jimenez et al., 2009; Ng et al., 2016). Consequently, these
results underscore the need for SOA models to incorporate fragmentation-dominated pathways to accurately
represent wintertime haze evolution.
**4 Conclusions**
This study provides a comprehensive characterization of wintertime submicron aerosols ($PM_1$) in Seoul, integrating
chemical composition, volatility measurements, and source apportionment to reveal critical insights into urban OA
evolution. The two most significant findings are the robust real-time identification of a nitrogen-containing organic
aerosol (NOA) factor and the observation of unexpected volatility behavior in highly oxidized OA. The NOA
factor, spectrally dominated by low-molecular-weight alkylamine fragments, was successfully resolved primarily
due to the accumulation of pollutants during wintertime stagnation, which sufficiently enhanced the spectral signals
of these semi-volatile species for identification. Its temporal and chemical characteristics point to a mixed
primary/secondary origin: driven by direct combustion emissions (e.g., residential heating) but significantly
enhanced by the rapid gas-to-particle partitioning of semi-volatile amines under cold, humid conditions.
Concurrently, the volatility analysis revealed a notable decoupling between oxidation state and volatility for the
More-Oxidized Oxygenated OA (MO-OOA). Despite its high O:C ratio (~1.15), MO-OOA exhibited elevated
volatility, a deviation from classical aging models that typically associate high oxidation with low volatility. This
behavior is attributed to the specific conditions of winter haze—reduced photolysis and high aerosol mass
loadings—which favor fragmentation-dominated aging pathways and the absorptive partitioning of volatile
oxygenated products.
These results revise our understanding of wintertime aerosol dynamics and underscore the limitations of current
models in representing reduced-nitrogen species and non-canonical oxidation pathways. To address the remaining
uncertainties, future research should prioritize evaluating the seasonal variability of NOA to better disentangle the
influence of meteorological drivers from specific emission sources. Concurrently, there is a critical need to directly
probe radical oxidation mechanisms, such as $RO_2$ autoxidation and $NO_3$ chemistry, particularly under haze
conditions. Integrating these field inquiries with laboratory studies and advanced molecular-level measurements
(e.g., FIGAERO-CIMS, EESI-TOF) will be essential for constraining the formation, lifetime, and climate impacts
of these complex organic aerosol components in polluted megacities.
**Data availability.**
Data presented in this article are available upon request to the corresponding author.
**Acknowledgements**
This work was supported by the National Research Foundation of Korea (NRF) grant funded by the Korea
government (MSIT) (RS-2025-00514570), the project "development of SMaRT based aerosol measurement and
analysis systems for the evaluation of climate change and health risk assessment" operated by Seoul National
University (900-20240101). Also this research was supported by Particulate Matter Management Specialized Graduate
Program through the Korea Environmental Industry & Technology Institute (KEITI) funded by the Ministry of Environment
(MOE).
**Author Contributions**
HK designed the study and prepared the manuscript. JJ operated the TD-AMS and analyzed the TD-AMS data. JM
curated and managed the dataset. HGK conducted the volatility analysis of organic aerosol (OA)

**Competing interests.**
The authors declare that they have no conflict of interest.
















**Tables and Figures**



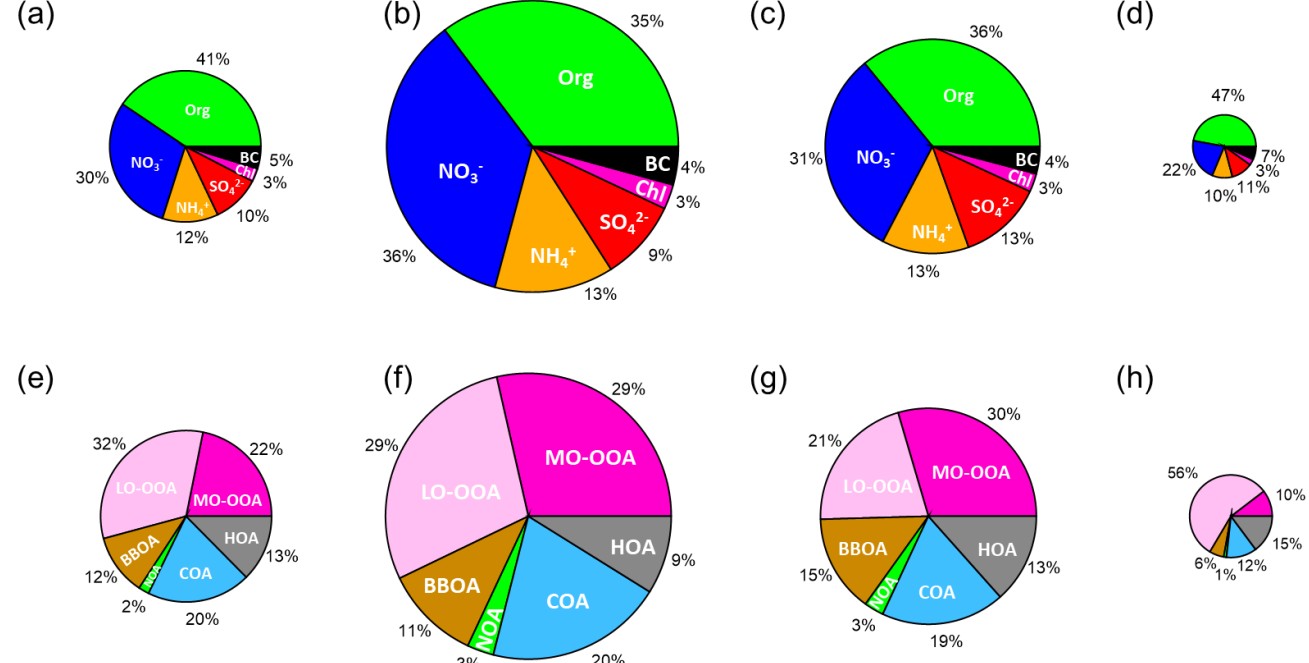


| | Period | Standard | Avg. Mass conc.( μg m⁻³) |
|---|---|---|---|
| **Total** | 2019.11.28 ~ 2019.12.28 | | Avg PM₁ = 26.37 |
| **Clean** | 2019.12.04 ~ 2019.12.06 | Daily PM₁ < 10.00 μg m⁻³ | Avg PM₁ = 9.98 |
| **Haze 1** | 2019.12.07 ~ 2019.12.11 | Daily PM₁ > 30.00 μg m⁻³ | Avg PM₁ = 51.88 |
| **Haze 2** | 2019.12.21 ~ 2019.12.25 | Daily PM₁ > 30.00 μg m⁻³ | Avg PM₁ = 37.71 |


**Figure 1.** Compositional pie charts of PM₁ species for (a) the entire study period, (b) haze period 1, (c) haze period 2, and (d) a clean period; and of each OA source for (e) the entire study period, (f) haze period 1, (g) haze period 2, and (h) the clean period.Table. Standard and average PM₁ mass concentrations during the entire study period, haze period 1, haze period 2, and the clean period.

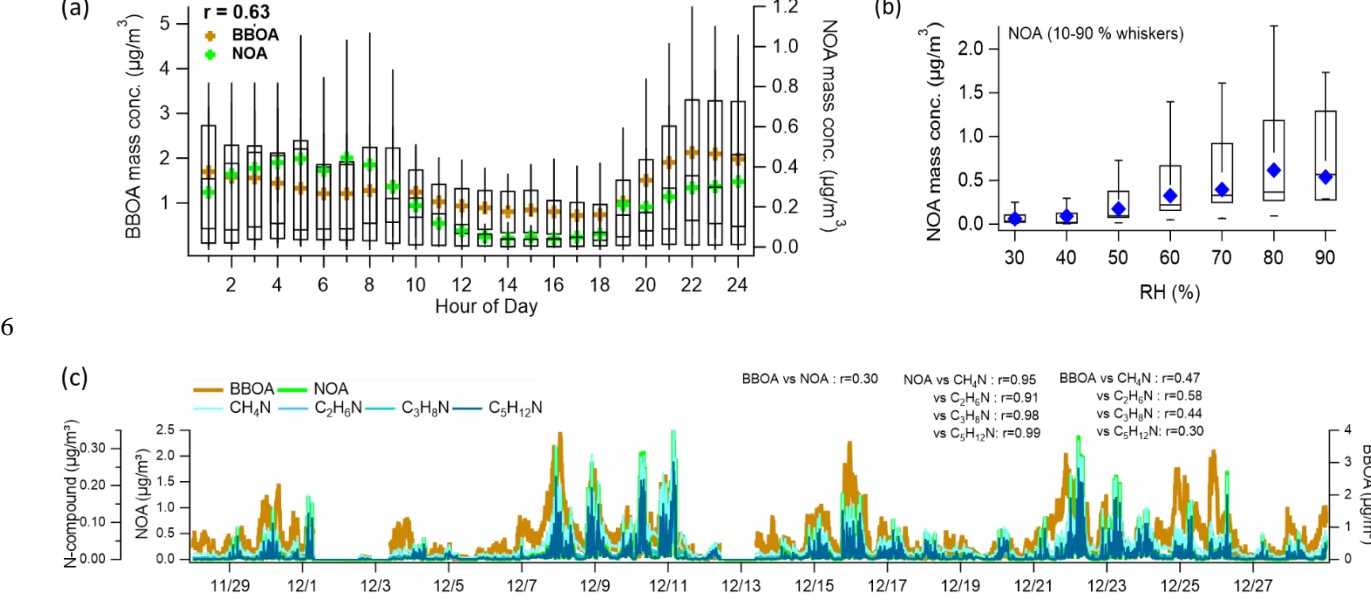



**Figure 2.** (a) Diurnal mean profiles of NOA and BBOA. Whiskers denote the 90th and 10th percentiles; box
edges represent the 75th and 25th percentiles; the horizontal line indicates the median, and the colored marker
shows the mean. The diurnal correlation between NOA and BBOA mean values is 0.63.
(b) Relative humidity (RH)-binned nighttime (19:00–05:00) profile of NOA. Box and whisker definitions are the
same as in panel (a). (c) Time series of NOA, BBOA, and amine-related ions ($CH_4N^+$, $C_2H_6N^+$, $C_3H_8N^+$,
$C_5H_{12}N^+$), along with their correlations with NOA and BBOA.

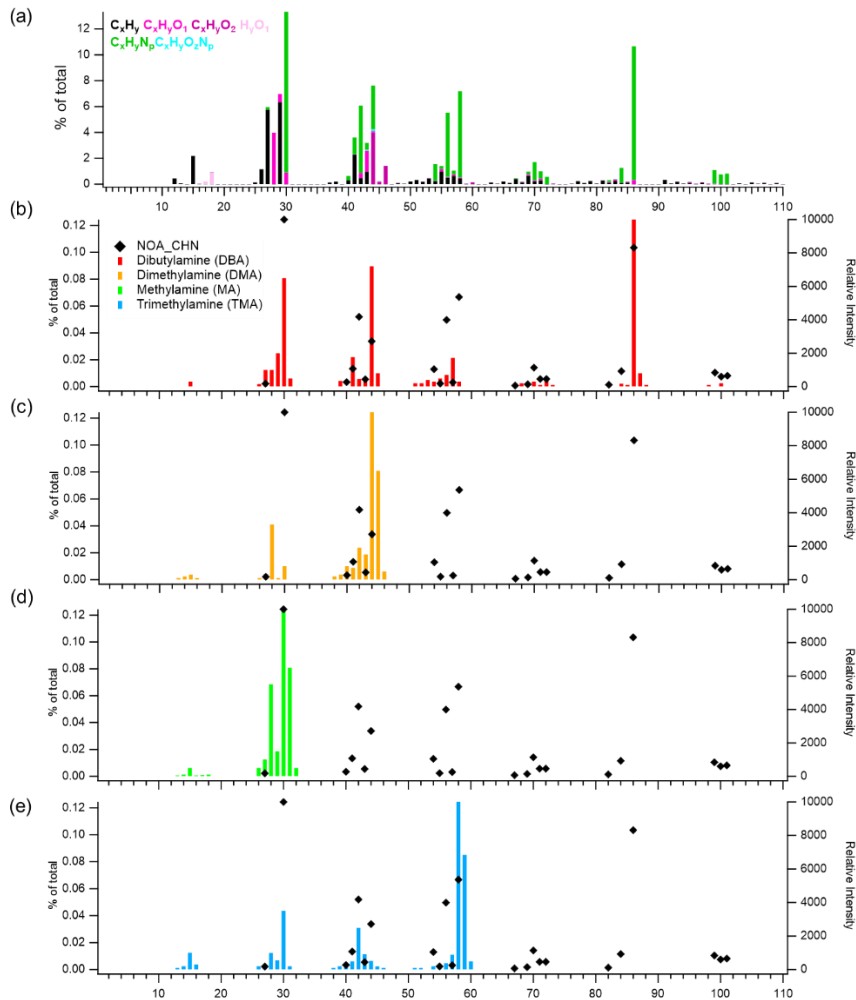

**Figure 3.** Mass spectra of (a) the NOA factor resolved by PMF analysis in this study, and reference spectra of amines from
the NIST library: (b) dibutylamine (DBA), (c) dimethylamine (DMA), (d) methylamine (MA), and (e) trimethylamine
(TMA). In panels (b)–(e), the left y-axis indicates the contribution of CHN-containing ions in the NOA factor (% of total),
while the right y-axis shows the relative intensity of each compound's mass spectrum from the NIST library.

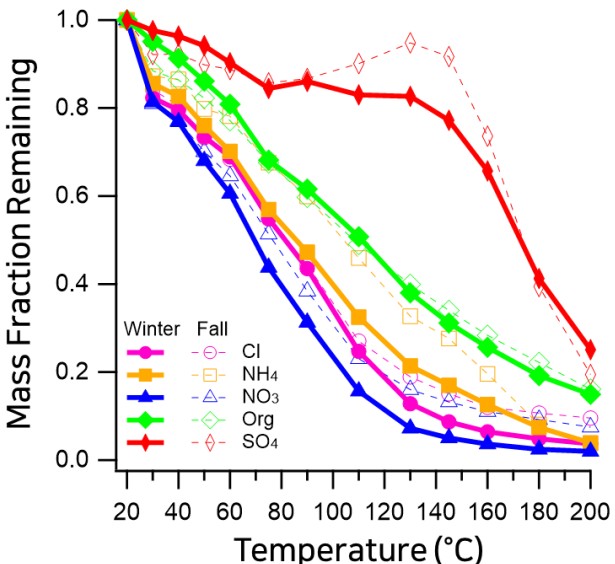

**Figure 4.** Mass fraction remaining (MFR) of non-refractory (NR) aerosol species measured in Seoul using a thermodenuder
coupled to a high-resolution time-of-flight aerosol mass spectrometer (HR-ToF-AMS).  Winter 2019 (this study; dashed) is
compared with fall 2019 (previously reported; solid) (Jeon et al., 2023).Species include organics (magenta), nitrate (blue),
sulfate (orange), ammonium (green), and chloride (red).

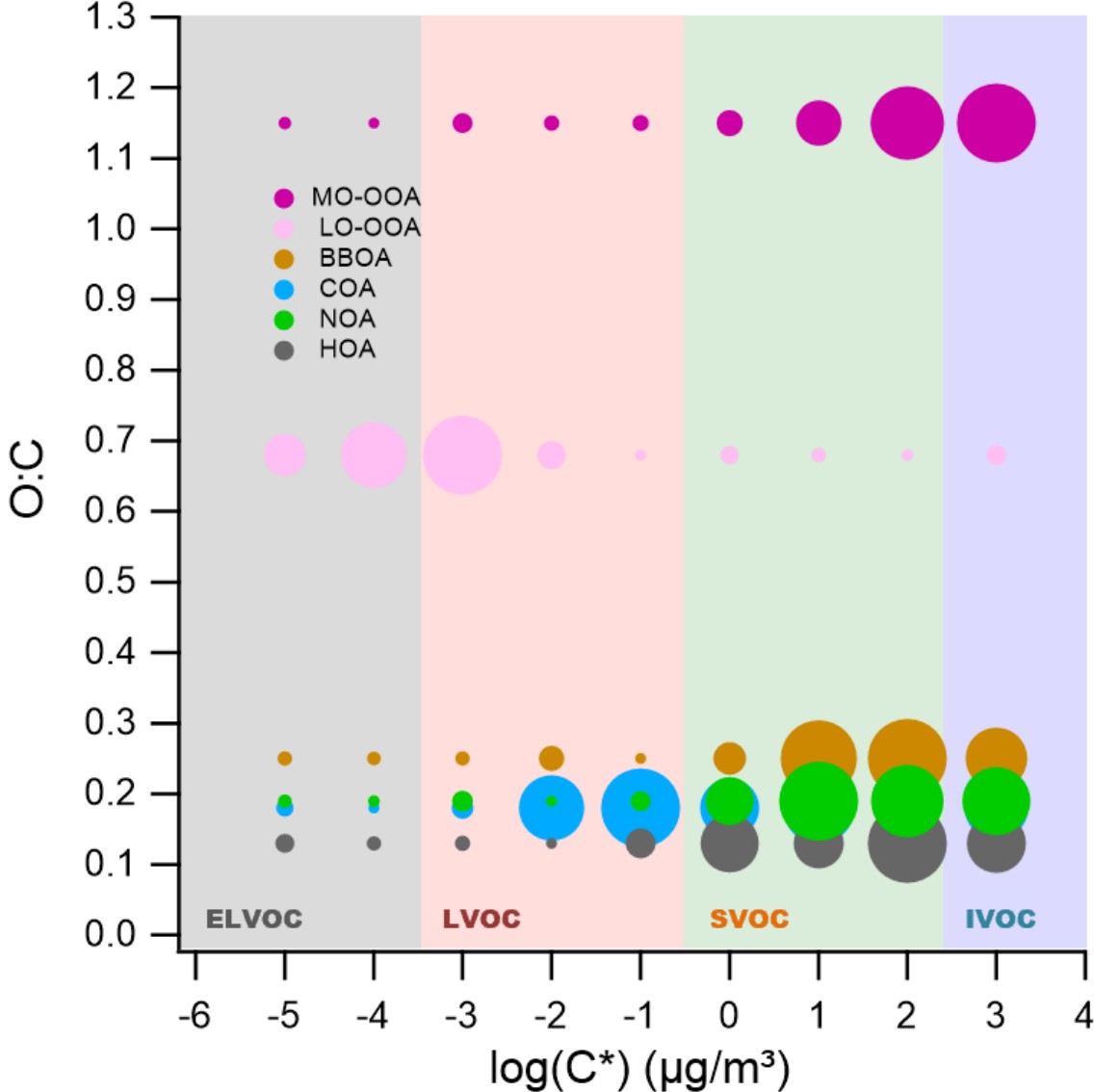

**Figure 5.** Two-dimensional volatility basis set (2D-VBS) representation of organic aerosol (OA) sources identified in winter 2019 in Seoul. The plot illustrates the relationship between the oxygen-to-carbon (O:C) ratio and the effective saturation concentration (C*) for each OA source resolved via positive matrix factorization (PMF). Solid circles represent the volatility distribution across C* bins, with marker size proportional to the mass fraction within each bin for the given source. Shaded regions correspond to different volatility classes: extremely low-volatility organic compounds (ELVOCs), low-volatility organic compounds (LVOCs), semi-volatile organic compounds (SVOCs), and intermediate-volatility organic compounds (IVOCs), delineated by their C* values.

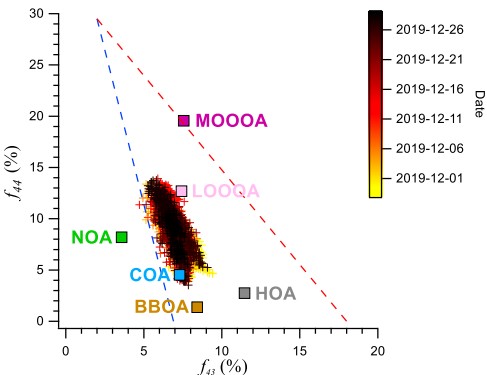

**Figure 6.** scatterplot of $f_{44}$ ($CO_2^+$) versus $f_{43}$ ($C_2H_3O^+$). for the measured organic aerosol. The data points are color-coded by
date to illustrate the temporal variation in OA composition throughout the observation period. The separated OA factors
(HOA, COA, BBOA, NOA, LO-OOA, and MO-OOA) are also shown to enable comparison of source contributions and
oxidation characteristics. The dashed line represents the typical $f_{60}$ threshold associated with biomass-burning influence,
while the triangular boundary indicates the conventional oxidative aging trend in the $f_{44}$–$f_{60}$ space.

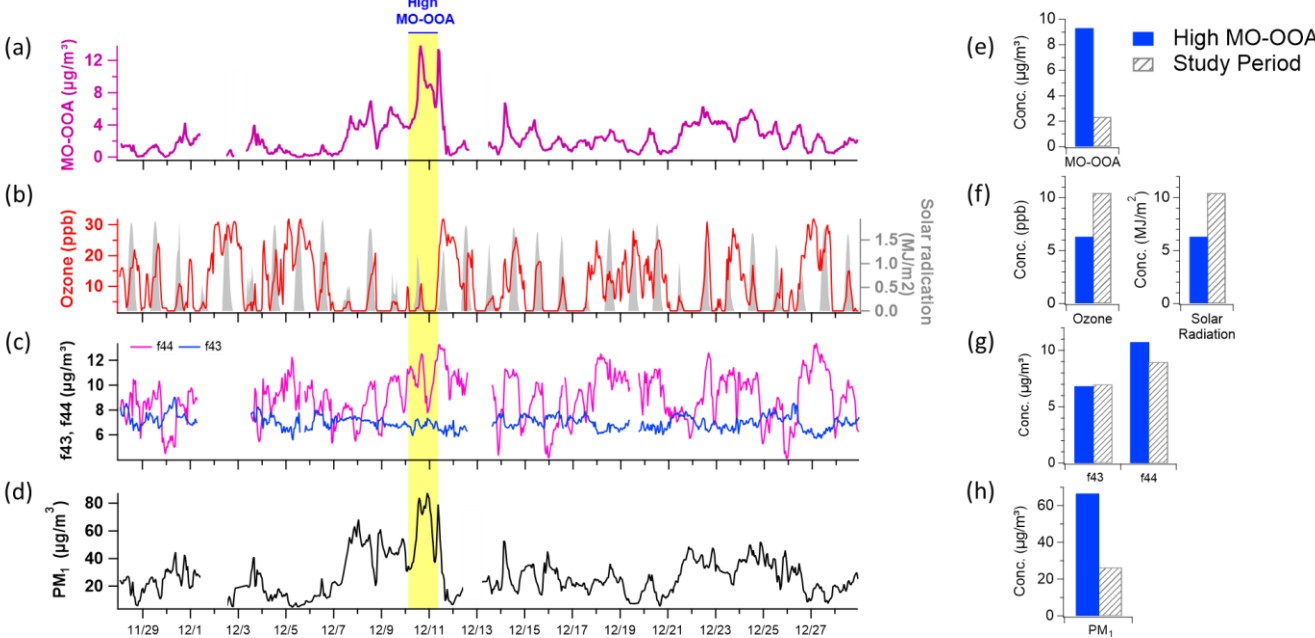

**Figure 7.** Time series plots of (a) MO-OOA concentration, (b) ozone (O₃) and solar radiation, (c) f₄₄ and f₄₃ (indicative of oxidation state), and (d) total PM₁ concentration. The period characterized by elevated MO-OOA levels is highlighted in bright yellow. Panels (e)–(f) present comparative distributions of these variables—MO-OOA, O₃ and solar radiation, f₄₄ and f₄₃, and PM₁—between the high MO-OOA period (shaded in blue) and the entire measurement period (indicated by gray hatching).

532

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
