# Peer review of "Source-Resolved Volatility and Oxidation State Decoupling in"

_EGUsphere, 2025_

## Author Comment (AC2)

**Response to Reviewer # 1**

We appreciate the anonymous reviewer for the thoughtful reviews and comments. We have carefully considered the suggestions and revised the manuscript accordingly. The reviewer comments are in blue, our comments are in black, and modifications to the manuscript are in red.

**General comments:**

Kim et al. investigated the characteristics of organic aerosol (OA) in Seoul during wintertime haze events. In contrast to previous studies conducted in the region, this work successfully identified a nitrogen-containing OA (NOA) factor, likely associated with biomass burning, in addition to five other OA factors using positive matrix factorization (PMF). A notable finding was that the oxidation state of oxygenated OA (OOA) did not align with volatility: the less-oxidized OOA (LO-OOA) was found to be less volatile than the more-oxidized OOA (MO-OOA). This observation challenges the conventional classification of OOA as semi-volatile or low-volatile and supports the use of LO-OOA and MO-OOA terminology.

The study provides insights relevant to understanding air quality in Seoul, one of the world's largest metropolitan areas, where haze events are increasingly linked to nitrogen-related pollutants. Although NOA contributed only a minor fraction of total OA, its enhancement offers clues about particle-phase chemistry that may worsen air quality and could inform the design of mitigation strategies. Furthermore, this study represents the first attempt in Seoul to connect the oxidation state of OA with volatility, adding useful information to the literature.

However, many of the findings presented are not conceptually new within the broader atmospheric chemistry field. In addition, several interpretations remain speculative, with limited supporting evidence. In particular, Section 3.3 does not convincingly explain why MO-OOA in Seoul appears more volatile than LO-OOA. For these reasons, the work may be more appropriate for publication as a Measurement Report in its current form, unless the discussion and interpretation are substantially strengthened.

Thank you for the thoughtful assessment and for recognizing the potential of our study. We appreciate the concern that some interpretations initially appeared speculative. Guided by your comments, we have substantially strengthened Section 3.3 and clarified several points across the manuscript. We believe the revised paper now goes beyond a measurement-only contribution and meets the criteria for a research article in ACP. In addition to reporting winter measurements, the revision explains the chemistry, evaluates alternative explanations, and situates Seoul in comparison with other major cities—consistent with ACP research-article standards rather than a Measurement Report. We believe these advances are appropriate for a research article in ACP.

Additional comments are provided below.

**Major comments:**

Section 3.2: This study is presented as a characterization of wintertime aerosol, yet comparisons are made with fall data. Please clarify the source of the fall dataset—was it obtained from your previous work or from another published study? If it is adopted, this needs to be explicitly stated earlier in the section and consistently noted throughout the manuscript, including in figure captions.

Thank you for the comment. Section 3.2 analyzes **only** the winter-2019 dataset collected in this study. To avoid ambiguity, we clarified the phrasing at the few places where this could be misread.

1. **Line 258**: "Compared to fall (T50 ~73 °C, incomplete evaporation), winter nitrate appeared more volatile, supporting enhanced NOA detection and indicating relatively fewer non-volatile nitrate forms." After: "Compared to previously reported fall conditions (T50 ~73 °C, incomplete

- evaporation), winter nitrate appeared more volatile, supporting enhanced NOA detection and indicating relatively fewer non-volatile nitrate forms (e.g., Kang et al., 2022; Jeon et al., 2023)."
- 2. **Line 266:** "Residual ammonium at 200 °C was consistent (~4%) across seasons (Kang et al., 2022; Jeon et al., 2023)." **After:** "Residual ammonium at 200 °C was consistent (~4%) in previously reported spring/fall measurements (Kang et al., 2022; Jeon et al., 2023)."
- 3. Line 268: "Chloride volatility was also comparable between seasons in terms of T50, but exhibited more complete evaporation in winter (~4% residual vs. ~10% in fall), possibly reflecting a shift in source to more volatile forms like road salt during wintertime." After: "Chloride volatility was also comparable across seasons in prior studies in terms of T50, but exhibited more complete evaporation in winter (~4% residual vs. ~10% in fall), possibly reflecting a shift in source to more volatile forms like road salt during wintertime."
- 4. **Line 274**: "This trend aligns with spring and fall observations (Kang et al., 2022; Jeon et al., 2023)." **After:** "This trend aligns with previously reported spring and fall observations (Kang et al., 2022; Jeon et al., 2023)."
- 5. Line 292: "This pattern reflects its diverse cooking sources and variable emission profiles (Kang et al., 2022)."After: "This pattern reflects its diverse cooking sources and variable emission profiles as previously reported (Kang et al., 2022)."
- 6. **Figure 4 caption :** "Mass fraction remaining (MFR) of non-refractory (NR) aerosol species measured in Seoul using a thermodenuder coupled to a high-resolution time-of-flight aerosol mass spectrometer (HR-ToF-AMS). Winter 2019 (this study; dashed) is compared with fall 2019 (previously reported; solid) (Jeon et al., 2023). Species include organics (magenta), nitrate (blue), sulfate (orange), ammonium (green), and chloride (red).

Section 3.3: The discussion in this section could be substantially strengthened. For instance, the relatively high volatility of MO-OOA may represent a distinct feature of Seoul compared to other megacities. A more thorough comparison with results from other TD-AMS studies would help contextualize your findings. Given that one of the stated objectives of this work is to "improve understanding of wintertime OA in Seoul," highlighting how Seoul's OOA differs from or aligns with other urban environments would significantly enhance the discussion.

Thank you. Per your suggestion we've strengthened section 3.3 with explicit TD-AMS comparisons to other megacities in the discussion of section 3.3. The updated paragraph (below) now cites representative studies from Mexico City/Los Angeles, Paris, Beijing, and Shenzhen, plus springtime Seoul for seasonal context.

"Viewed against prior TD-AMS results, the volatility of Seoul's winter MO-OOA presents a unique case, particularly in the nature of its O:C-volatility relationship. Prior urban studies have commonly reported substantial SVOC-OA, consistent with high photochemical activity or elevated loadings; for example, Mexico City/Los Angeles showed pronounced SVOC–IVOC contributions during warm seasons (Cappa and Jimenez, 2010), and summertime Beijing and wintertime Shenzhen likewise exhibited strong overall OA volatility (Xu et al., 2019; Cao et al., 2018). While these comparisons establish that volatile OA is common, they generally did not report the factor-level inversion observed here, where the highly-oxidized OOA component (MO-OOA) was *more* volatile than a less-oxidized OOA (LO-OOA). This behavior is distinct from findings in colder, lower-loading regimes; wintertime Paris, for instance, maintained the conventional hierarchy where the more-oxidized OOA was comparatively less volatile (Paciga et al., 2016). Furthermore, seasonal context within Seoul showed springtime OA with lower oxidation levels than our winter MO-OOA despite similar SVOC contributions (Kang et al., 2022). This comprehensive comparison underscores the unusual nature of the O:C-volatility relationship observed under the specific winter haze conditions in Seoul.

Line 163: The claim that enhanced MO-OOA and NOA formation during haze is due to stagnation is not convincing without supporting meteorological data. Please provide wind speed/direction or back-trajectory analyses.

Thank you for this comment. We now document meteorological support for stagnation during haze by (i) presenting back-trajectory clustering that identifies a local recirculation regime (Cluster 1; Fig. S8) and (ii) showing persistently low wind speeds during haze windows in the campaign time series (Fig. S6). These diagnostics indicate that the haze episodes were dominated by local accumulation rather than long-range transport, consistent with the observed co-enhancement of MO-OOA and NOA. Now the relevant section reads:

"Back-trajectory clustering shows frequent short-range recirculation over the Seoul Metropolitan Area during haze (Cluster 1; Fig. S8), and the time series indicates persistently low surface wind speeds during these periods  $(1.73 \pm 0.89 \text{ vs. } 2.34 \pm 1.18 \text{ (clean)})$  (Fig. S6), together pointing to stagnation-driven accumulation of local emissions; the concurrent increases in MO-OOA and NOA are therefore consistent with enhanced in-city formation under stagnant conditions."

Line 172: Could factor resolution differ between V- and W-mode? Please show whether NOA is detected in both modes and perform PMF sensitivity tests to mode/resolution.

Thank you for the careful question. In the HR-ToF-AMS, W-mode provides higher mass resolving power than V-mode, enabling separation of near-isobaric ions (e.g., m/z 58, 86, 100 series) that are important for resolving amine-rich (NOA) spectra; V-mode offers higher sensitivity but lower resolution and is widely used for quantification and bulk metrics, whereas W-mode is preferred for factor analysis of HR ion families. Consistent with standard AMS practice, our PMF was performed on W-mode HR spectra (Methods), where the amine fragments are better separated and uncertainties are well characterized. Running PMF on the lower-resolution V-mode would merge key N-containing ions with interferences and degrade factor identifiability, so we did not perform a V-mode PMF.

Line 194: Since amines can also originate from solvents, might some portion of NOA be linked to solvent emissions? Given Seoul's solvent usage, this possibility should be addressed.

Thank you for the helpful comment. We agree that solvent- and industry-related amines (e.g., tertiary amines used in polyurethane manufacturing/foams, metalworking, printing, and wastewater treatment) can contribute to urban amine budgets. Reviews and technical sources document such emissions and uses (e.g., TEA/DEEA/DMEA in PU catalysts), and Korean inventories indicate that solvent use is a major component of anthropogenic VOC emissions nationally(references). However, In our dataset, several diagnostics favor near-source combustion/BB influences for NOA during winter in Seoul: (i) night—early-morning peaks and co-variation with BBOA, (ii) back-trajectory clusters indicating local recirculation, and (iii) mass-spectral features dominated by small alkylamine fragments (e.g., m/z 30, 44, 58, 86) that match low-molecular-weight alkylamines commonly associated with combustion/BB rather than tertiary amines typical of PU/solvent use. That said, we now explicitly note that a minor contribution from solvent/industrial amines cannot be ruled out, and we added references and a sentence in the Discussion to reflect this possibility.

"These amines are commonly emitted during the combustion of nitrogen-rich biomass and proteinaceous materials and are frequently associated with biomass-burning emissions (Ge et al., 2011). Previous molecular analyses in Seoul also indicate DMA, MA, and TMA as the dominant amine species in December

(Baek et al., 2022). While other amines such as triethylamine (TEA), diethylamine (DEA), and ethylamine (EA) may contribute via industrial/solvent pathways (e.g., chemical manufacturing, petrochemical corridors, wastewater treatment), our HR-AMS spectra are dominated by small alkylamine fragments (m/z 30, 44, 58, 86) and the diurnal behavior co-varies with combustion markers (below), indicating a primarily combustion-linked influence. Nevertheless, recent urban measurements and sector-based analyses show that industrial activities can contribute measurable amines in cities (Tiszenkel et al., 2024; Zheng et al., 2015; Mao et al., 2018; Shen et al., 2017; Yao et al., 2016). Accordingly, a minor NOA contribution from solvent/industrial amines cannot be excluded."

Line 199: Where do you expect biomass burning that could influence Seoul's air to occur? Since NOA (linked to BBOA) is a central finding, more discussion on plausible sources and transport pathways would be valuable.

Thank you for this valuable suggestion. We expanded the discussion to identify plausible biomass-burning (BB) source regions and pathways that could influence Seoul and to clarify why our dataset points primarily to local influences. Specifically: (i) wintertime urban/peri-urban small-scale BB (residential solid-fuel use, restaurant charcoal use, and intermittent waste burning) has been observed in Seoul and surroundings (Kim et al., 2017); (ii) agricultural residue burning occurs seasonally in nearby provinces and can episodically affect the Seoul Metropolitan Area (Han et al., 2022); and (iii) regional transport from upwind regions (including northeastern China/North Korea) can influence Korea under northerly/northwesterly flow (Lamb et al., 2018; Nault et al., 2018). In our case, the night—early-morning peaks, co-variation with BBOA, and back-trajectory clusters indicating local recirculation are most consistent with near-source emissions within the metropolitan area during the study period (Yoo et al., 2024). We added further discussion at the end of section 3.1.3.

"Regarding source location, several pathways can influence Seoul's biomass burning signature. First, urban/peri-urban small-scale burning (e.g., solid-fuel use in select households, restaurant charcoal use, and intermittent waste burning) has been reported and can enhance BBOA locally (Kim et al., 2017). Second, nearby agricultural-residue burning in surrounding provinces occurs seasonally and can episodically impact the metropolitan area (Han et al., 2022). Third, regional transport from upwind regions (e.g., northeastern China/North Korea) can bring biomass burning influenced air masses under northerly/northwesterly flow (Lamb et al., 2018; Nault et al., 2018). In this dataset, the nighttime and early-morning enhancements, the BBOA–NOA co-variation, and trajectory clusters showing regional recirculation indicate a predominantly local/near-source contribution during the study period (Yoo et al., 2024), with episodic non-local influences remaining possible."

Line 225: The phrase "...with inorganic secondary species such as biomass burning" is unclear. Please revise.

Thank you for pointing this out. We agree the phrase was ambiguous—biomass burning is a source, not an inorganic secondary species. We removed the confusing wording and clarified the comparison to sulfate, nitrate, and ammonium. Now the relevant line reads:

"In contrast, LO-OOA exhibited only modest correlations with sulfate, nitrate, and ammonium (r = 0.50, 0.51, and 0.42, respectively), suggesting additional contributions from semi-primary sources not closely linked to inorganic secondary formation (e.g., cooking, traffic, biomass burning)."

Line 226: Earlier you noted that LO-OOA lacks m/z 60, but here you attribute it partly to combustion-related activities. This appears contradictory. Please clarify or rewrite.

Thank you for noting the ambiguity. We agree that invoking m/z 60 (levoglucosan) alongside an earlier statement that LO-OOA lacks m/z 60 is confusing. Now the section reads:

"LO-OOA does not exhibit a pronounced m/z 60 (levoglucosan) signal (Fig. S2); however, the levoglucosan marker (f60) is known to diminish with atmospheric aging and can become weak or undetectable downwind (Hennigan et al., 2010; Cubison et al., 2011). Taken together, the weaker coupling with secondary inorganics and the absence of a strong m/z 60 peak indicate that LO-OOA is a mixture of aged secondary organics and semi-primary urban emissions, while a contribution from aged biomass-burning influence cannot be ruled out."

Line 257–258: The text seems to imply that NOA could be detected as nitrate. However, most reduced nitrogen species (other than nitro-aromatics, based on Xu et al., 2021, AMT) are unlikely to be detected as nitrate by AMS. Please clarify.

Thank you for pointing this out. We agree that reduced-nitrogen species (e.g., amines) are not detected by the AMS as "nitrate." In the AMS, the nitrate channel reflects inorganic nitrate and organic nitrates (RONO2) via NO+/NO2+ fragments; most reduced-N compounds do not contribute, with the noted exception that nitroaromatics can appear in the nitrate channel. To avoid confusion, we removed NOA from this sentence and now discuss nitrate volatility independently.

Compared to previously reported fall conditions ( $T_{50} \sim 73$  °C, incomplete evaporation), winter nitrate appeared more volatile, indicating relatively fewer non-volatile nitrate forms (e.g., Kang et al., 2022; Jeon et al., 2023).

Line 261: If metallic sulfates contribute to PM1, then defining PM1 as NR-PM1 + BC may underestimate total mass. Please comment. Thank you for the helpful comment. Metallic sulfates (e.g., CaSO4, MgSO4, Na2SO4) would not be fully captured by the AMS and, if abundant in PM1, could bias PM1 = NR-PM1 + BC low. However, for our winter urban period in Seoul:

- 1. The HR-ToF-AMS measures non-refractory submicron species; refractory salts typically do not flash-vaporize at the AMS vaporizer and contribute negligibly to the AMS sulfate signal (Allan et al., 2004; Canagaratna et al., 2007).
- 2. Multiple Seoul/Korea studies show PM1 is dominated by NR species + BC, with sea-salt/crustal components largely in the coarse mode; our analysis window also excluded dust events, so refractory PM1 was minimal .
- 3. The subtle ~140 °C slope change is more plausibly due to ammonium-sulfate system behavior (e.g., morphology/phase-state changes in (NH4)2SO4/NH4HSO4 and mixed organic–inorganic particles) and/or organosulfate mixing, rather than metallic sulfates (Huffman et al., 2009; Faulhaber et al., 2009).

Accordingly, using  $PM_1 = NR-PM_1 + BC$  does not materially underestimate  $PM_1$  for this dataset. We added a sentence in Methods clarifying the non-refractory definition and a note in Results that we interpret the ~140 °C feature as ammonium-sulfate/phase-state and/or organosulfate mixing—not metallic sulfate. Now the relevant section reads;

Line 261: "Sulfate was the least volatile ( $T_{50} \approx 170$  °C), consistent with ammonium sulfate (Scott and Cattell, 1979). A subtle slope change near 140 °C likely reflects ammonium-sulfate morphology/phase-state changes and/or organosulfate—inorganic mixing, rather than contributions from metallic (refractory) sulfates, which are not efficiently detected by AMS."

Methods: "PM1 mass in this study is taken as NR-PM1 (from AMS) + black carbon (BC; measured by MAAP), which is appropriate for winter Seoul where refractory PM1 (metal/sea-salt/crustal) is minor and dust events were excluded (e.g., Kim et al., 2017; Nault et al., 2018; Kang et al., 2022; Jeon et al., 2023)."

Line 289: Since MO-OOA is described as more volatile, the phrase "consistent with its aged, highly condensed nature" seems contradictory. Consider deleting or rephrasing.

Thank you for the suggestion. We deleted the phrase.

Line 308: What alternative oxidation mechanisms or environmental conditions could explain the observed inverse relationship between O:C ratio and volatility? Please specify and cite relevant studies.

Thank you for the comment. We agree the original lead-in was too vague on the mechanisms. We revised the sentence to explicitly foreshadow the explanation that immediately follows (autoxidation-driven fragmentation under low-OH/low- $O_3$  haze conditions and enhanced condensation at high particle loadings). We also added a section pointer for clarity.

"This observation appears to contradict the usual inverse O:C-volatility relationship; however, under winter haze conditions—with suppressed O3/low OH, particle-phase autoxidation and fragmentation can yield higher-O:C yet more volatile products, with enhanced condensation on abundant particle surface area (details below)."

Line 311: The authors attribute reduced OH to suppressed O3 photolysis under haze. However, in polluted boundary-layer conditions, OH production often depends strongly on HONO photolysis in addition to O3 photolysis. If the authors want to keep this statement, I suggest add discussion and relevant reference on this.

Thank you for the comment. We did not measure HONO directly, so our "low-OH" interpretation was based primarily on reduced  $O_3$  photolysis during haze. We now clarify that haze generally reduces actinic flux for *all* photolysis channels, including HONO photolysis, due to aerosol extinction/scattering. At the same time, HONO concentrations often increase at night during polluted, humid conditions because of heterogeneous  $NO_2 \rightarrow HONO$  production on surfaces/aerosols and direct/near-surface emissions; thus the net OH from HONO can remain significant or even episodically important despite lower j(HONO). We added text noting both effects and cited field studies in Korea (KORUS-AQ) and China that document (i) reduced photolysis frequencies under haze and (ii) enhanced HONO production via heterogeneous pathways. This balanced perspective does not change our main conclusion (a low-OH regime relative to sunny, high-photolysis periods), but it acknowledges the potential compensating role of HONO under haze.

"We note that haze also suppresses HONO photolysis; however, HONO concentrations can be elevated at night and early morning via heterogeneous NO2 conversion and surface emissions, so net OH from HONO may remain non-negligible even as photolysis rates are depressed (e.g., Gil et al., 2021; Kim et al., 2024; Slater et al., 2020).

Line 312: Under conditions of reduced OH, how are RO2 radicals formed and how might they contribute to particle-phase autoxidation? Please elaborate and provide references.

Thank you for the prompt. We clarified that even under reduced OH (low  $j(O^1D)$  during haze), RO2 can arise from (i) residual gas-phase ozonolysis of alkenes (Ehn et al., 2014; Ziemann and Atkinson, 2012; Bianchi et al., 2019), (ii) nighttime/low-light NO3-initiated chemistry where NOx and remaining O3 permit NO3 formation (Brown and Stutz, 2012; Ziemann and Atkinson, 2012), and (iii) condensed-phase radical chemistry/autoxidation in particles once peroxy radicals are initiated (Kroll and Seinfeld, 2015; Berndt et al., 2016; Bianchi et al., 2019). These routes can sustain RO2 pools that drive particle-phase autoxidation under haze. Now the line reads:

"Even under low-OH conditions, NO3 formed via NO2 + O3 can initiate RO2 production through addition to alkenes, while reduced photolysis at night/low light extends NO3 lifetimes; these RO2 then participate in particle-phase autoxidation, yielding highly oxygenated yet relatively volatile products (Brown and Stutz, 2012; Ziemann and Atkinson, 2012; Ehn et al., 2014; Berndt et al., 2016; Bianchi et al., 2019)."

Line 315–316: To support the interpretation that fragmentation contributes to the high volatility of MO-OOA, please cite previous studies that compared OA composition using both online and offline techniques.

Thank you for the comments. We now cite studies that combine online (AMS/FIGAERO-CIMS) and offline (filter HRMS/GC-MS) techniques showing that fragmentation-dominated aging yields high-O:C yet more volatile products and elevated f44 without proportional increases in f43.

"Consistent with this interpretation, online AMS/FIGAERO-CIMS and EESI-TOF, as well as offline HRMS/GC-MS, have reported high-O:C yet more-volatile product distributions accompanied by elevated  $f_{44}$  with comparatively stable  $f_{43}$  under fragmentation-dominated aging (Kroll et al., 2009; Ng et al., 2010; Chhabra et al., 2011; Lambe et al., 2012; Lopez-Hilfiker et al., 2016; D'Ambro et al., 2017)."

Line 317: Please revise "semi-volatile species" to "semi-volatile/intermediate-volatility organics." Additionally, please clearly comment that functionalized but low-molecular-weight compounds can fall in the SVOC–IVOC range and may contribute to the high volatility of MO-OOA. Please provide supporting references.

We revised "semi-volatile species" to "semi-volatile/intermediate-volatility organics (SVOC–IVOC)" and explicitly note that functionalized, low-molecular-weight compounds can fall in the SVOC–IVOC range and thus contribute to the high apparent volatility of MO-OOA. References added.

"Furthermore, high aerosol mass loadings during haze events provide abundant surface area for the uptake of semi-volatile/intermediate-volatility organics (SVOC–IVOC) via absorptive partitioning, so that higher COA enhances condensation (Pankow, 1994; Donahue et al., 2006; Hallquist et al., 2009; Robinson et al., 2007). We also note that functionalized, low–molecular-weight compounds can reside in the SVOC–IVOC range and thus contribute to the high apparent volatility of MO-OOA (Ng et al., 2010; Chhabra et al., 2011; Lopez-Hilfiker et al., 2016; D'Ambro et al., 2017)."

Line 327–328: A comparison of average OA mass spectra between haze and non-haze periods would be informative. Do non-haze periods show relatively more high-molecular-weight fragments?

Thank you for the insightful comment. We added a direct comparison of average OA mass spectra for haze and non-haze periods (new Fig. S12; events defined in Fig. S6). Because AMS EI at 70 eV produces extensive fragmentation, non-haze periods do **not** exhibit "more high-molecular-weight fragments." The differences are expressed in low-m/z markers: haze spectra show enhanced oxygenated fragments (m/z 28 = CO+, 29 = CHO+, 44 = CO2+) together with higher  $f_{44}$  and O:C, while  $f_{43}$  remains comparatively unchanged. Non-haze spectra display larger fractional contributions of hydrocarbon-like ions (m/z 41, 43, 55, 57) and lower  $f_{44}$  and O:C. This pattern—increased  $f_{44}$  with flat  $f_{43}$ —is the canonical signature of fragmentation-dominated aging rather than functionalization (Kroll et al., 2009), reinforcing our interpretation that the periods of elevated MO-OOA reflect advanced oxidation and increased volatility arising from autoxidation and the condensation of small oxygenated fragments. Now the sentence reads;

"Consistent with this, the haze—non-haze comparison, including the high-MO-OOA interval (Fig. S12), shows larger oxygenated fragments (m/z 28, 29, 44) and higher f44 and O:C during haze, whereas non-haze periods exhibit relatively larger fractional hydrocarbon fragments (m/z 41, 43, 55, 57). These spectral contrasts indicate that the elevated volatility of MO-OOA reflects advanced oxidation—via autoxidation and the condensation of small oxygenated fragments—rather than enrichment of high-molecular-weight ions, particularly under conditions of limited OH and high particulate surface area."

**Specific comments:**

**2 Experimental methods**

Line 97: Since you are presenting quantitative results, please specify what collection efficiency (CE) value was applied in the AMS data analysis. In addition, could the use of the thermal denuder (TD) influence CE, for example by altering particle phase state or mixing characteristics? Please clarify whether you assumed the same CE with and without TD and provide justification or relevant references.

Thank you for this point. We apply time-resolved, composition-dependent CE(t) (Middlebrook et al., 2012) separately to the bypass and TD lines. This approach allows CE to vary with the evolving nitrate fraction and neutralization, which TD heating can influence. The campaign-average CE values were TD:  $0.55 \pm 0.08$  and bypass:  $0.53 \pm 0.04$ ; the mean difference is 0.02 (~3.7%), which is smaller than the combined uncertainty of the two estimates ( $\approx 0.09$ ). In other words, the CE(t) distributions are similar between lines (comparable means and standard deviations), and we did not force them to be equal—each line's CE(t) was used in its own mass quantification.

We agree that phase-state or mixing changes beyond composition could, in principle, perturb CE. However, prior TD-AMS studies report that such TD-induced CE effects are modest and predominantly multiplicative/near-uniform with respect to thermogram interpretation, not altering thermogram shapes or  $T_{50}$  ordering (Huffman et al., 2009; Faulhaber et al., 2009; Cappa & Jimenez, 2010). Given our use of CE(t) that already responds to composition changes, the small TD-bypass CE difference relative to uncertainty, and literature showing minor additional CE impacts from phase/bounce, we consider any residual CE effects beyond composition to be minor for the volatility metrics presented. We have clarified this in the Methods and added the supporting citations.

"NR-PM1 quantification followed established AMS protocols (Ulbrich et al., 2009; Zhang et al., 2011). Both the bypass and TD streams were processed using a time-resolved, composition-dependent collection efficiency CE(t) following Middlebrook et al. (2012). TD heating can modify particle water and phase state/mixing and thereby influence CE beyond composition (Huffman et al., 2009), but prior TD–AMS studies indicate that such effects are modest and largely multiplicative, which do not distort thermogram shapes or T50 ordering (Faulhaber et al., 2009; Cappa & Jimenez, 2010). In our data, the CE(t) statistics for the two lines were similar (campaign-average CE: TD =  $0.55 \pm 0.08$ ; bypass =  $0.53 \pm 0.04$ ;  $\Delta = 0.02 \approx 3.7\%$ , below the combined uncertainty  $\approx 0.09$ ). We therefore report volatility metrics with these line-specific CE(t) corrections applied and interpret potential residual CE effects as minor."

Finally, if you define PM1 mass as NR-PM + BC, it would be important to cite studies demonstrating that PM1 in Seoul contains only limited amounts of metals or other refractory components (e.g., dust, sea salt), otherwise this definition may underestimate the total PM1 mass.

Thank you for raising this point. In Seoul, multiple ambient studies indicate that submicron mass  $(PM_1)$  is dominated by non-refractory species (organics, sulfate, nitrate, ammonium) plus BC, with refractory metal, dust and sea salt contributing only a small fraction to  $PM_1$  (and mainly affecting supermicron/coarse modes). Our study period excludes dust events. Accordingly, defining  $PM_1 = NR-PM_1 + BC$  is consistent with prior Seoul observations and with regional aircraft/surface measurements during KORUS-AQ. We have added the following references and a brief note in Methods to clarify this assumption.

"PM1 mass in this study is taken as NR-PM1 (from AMS) + black carbon (BC; measured by MAAP), which is appropriate for winter Seoul where refractory PM1 (metal/sea-salt/crustal) is minor and dust events were excluded (e.g., Kim et al., 2017; Nault et al., 2018; Kang et al., 2022; Jeon et al., 2023).."

**3 Results and discussion**

Line 149: Please clarify why a PM level of  $\sim$ 28  $\mu$ g m-3 is considered "moderate." A reference or comparison to regional air quality standards would help.

Thank you for the suggestion. We now justify "moderate" using (i) prior winter PM1 in Seoul (27.5  $\mu$ g m-3 on average; stagnant episodes  $\approx$  44  $\mu$ g m-3), showing our mean (27.8  $\pm$  15.3  $\mu$ g m-3) is comparable to the historical winter mean and below stagnant loads, and (ii) a Korea-specific PM1/PM2.5  $\approx$  0.8, which implies an equivalent PM2.5  $\approx$  34.8  $\mu$ g m-3, near the national 24-h PM2.5 standard (35  $\mu$ g m-3). We have added these references and clarifications.

"We conducted continuous measurements from 28 November to 28 December 2019, characterizing a winter period with a mean PM1 concentration of  $27.8\pm15.3\mu\text{gm}-3$ . This concentration is characterized as moderate; it closely matches historical winter PM1 means in Seoul (Kim et al., 2017) and implies an equivalent PM2.5 concentration is about  $34.8\mu\text{gm}-3$  (using a Korea-specific PM1/PM2.5 $\approx$ 0.8 (Kwon et al., 2023), which is near the national 24-h PM2.5 standard ( $35\mu\text{gm}-3$ ) (AirKorea)."

Line 186: A reference regarding the atmospheric lifetime and reactivity of amines would strengthen this discussion.

Thank you for the helpful suggestion. We added citations documenting the short atmospheric lifetimes and high reactivity of alkyl amines due to fast gas-phase oxidation .

"Back-trajectory analysis linked these events to regional recirculation patterns (Cluster 1, Fig. S7), suggesting a predominantly local origin—consistent with the short atmospheric lifetimes and high reactivity of most amines (Ge et al., 2011; Nielsen et al., 2012; You et al., 2014; Hanson et al., 2011). "

Line 242: Please italicize all "m/z ##" and "f##" notations throughout the manuscript for consistency.

Thank you for pointing this out. We have applied consistent typography throughout the manuscript by italicizing all m/z and f notations (e.g., m/z 44, f44). This change has been implemented across the main text, figure and table captions, and the Supplementary Information to ensure uniform style.

Line 246: Instead of describing the correlation between NOA and BBOA as "moderate," please quantify the correlation coefficient and state whether it is higher relative to other factor pairs.

Thank you for the suggestion. We replaced the qualitative phrase "moderate" with quantitative values and clarified the context relative to other factor pairs. Specifically, the BBOA–NOA time-series correlation is r=0.30 (the highest among the NOA–factor pairs examined), and their diurnal-profile correlation is r=0.63. We retain the ion-level correlations for completeness.

"Quantitatively, BBOA and NOA correlate with r = 0.30 in the time series—the highest among the NOA–factor pairs in our dataset—and their diurnal profiles are more strongly correlated (r = 0.63). Consistent with this, nitrogen-containing ions characteristic of NOA,  $C_2H_4N^+$  (r = 0.67) and  $C_2H_6N^+$  (r = 0.56), co-vary with BBOA and are dominant peaks in the NOA mass spectrum (Fig. 2; Fig. S3; Fig. S10; Fig. S11)."

Line 248: The phrase "such as biomass burning" may be redundant here, since this point is already discussed in the BBOA section. Consider removing it.

Thank you for the suggestion. We agree that "such as biomass burning" is redundant given the discussion in the BBOA section. We have removed the phrase and streamlined the sentence for clarity. The sentence now reads:

"This overlap suggests a potential shared emission source or co-emission scenario, consistent with the coemission of both organic aerosols and reduced nitrogen-containing compounds"

**Line 300–303: Please provide supporting references for this discussion.**

We added references demonstrating that functionalization typically lowers volatility, while fragmentation increases it and can reduce SOA yields despite higher O:C; this supports our emphasis on pairing composition with direct volatility constraints. Now the section reads;

[revised manuscript text omitted]

- Paciga, A., Young, D. E., Ward, M. W., et al.: Volatility of organic aerosol and its components in the megacity of Paris, Atmos. Chem. Phys., 16, 2013–2031, https://doi.org/10.5194/acp-16-2013-2016, 2016. Pankow, J. F.: An absorption model of gas/particle partitioning of organic compounds in the atmosphere, Atmos. Environ., 28A, 185–188, 1994.
- Poste, A. E., Stroud, M. J., and Evans, J. V.: Particulate matter from wastewater treatment plants: a review, Water Environ. Res., 86, 313–322, https://doi.org/10.2175/106143014X14062131238612, 2014.
- Robinson, A. L., Donahue, N. M., Shrivastava, M. K., et al.: Rethinking organic aerosols: semivolatile emissions and photochemical aging, Science, 315, 1259–1262, https://doi.org/10.1126/science.1133061, 2007.
- Scott, W. D. and Cattell, F. C. R.: Vapor pressure of ammonium sulfates, Atmos. Environ., 13, 307–317, https://doi.org/10.1016/0004-6981(79)90225-6, 1979.
- Slater, E. J., Gkatzoflias, D., Wang, Y., et al.: Elevated levels of OH observed in haze events during wintertime Beijing, Atmos. Chem. Phys., 20, 14847–14871, https://doi.org/10.5194/acp-20-14847-2020, 2020.
- Sun, Y., Zhang, Q., Schwab, J. J., et al.: Characterization of the sources and properties of organic aerosol from AMS measurements during a winter campaign in Beijing, China, Atmos. Chem. Phys., 10, 8951–8971, https://doi.org/10.5194/acp-10-8951-2010, 2010.
- Sun, Y., Du, W., Wang, Q., et al.: Real-time characterization of aerosol chemistry and mixing state at a high-altitude site in the southeastern Tibetan Plateau, Atmos. Chem. Phys., 18, 12891–12908, https://doi.org/10.5194/acp-18-12891-2018, 2018.
- Tiszenkel, L., Flynn, J. H., and Lee, S.-H.: Measurement report: Urban ammonia and amines in Houston, Texas, Atmos. Chem. Phys., 24, 11351–11363, https://doi.org/10.5194/acp-24-11351-2024, 2024.
- Xu, W., Han, T., Du, W., et al.: Summertime aerosol volatility measurements in Beijing, Atmos. Chem. Phys., 19, 10205–10216, https://doi.org/10.5194/acp-19-10205-2019, 2019.
- Yao, L., Wang, M.-Y., Wang, X.-K., et al.: Detection of atmospheric gaseous amines and amides by a high-resolution time-of-flight chemical ionization mass spectrometer with protonated ethanol reagent ions, Atmos. Chem. Phys., 16, 14527–14543, https://doi.org/10.5194/acp-16-14527-2016, 2016.
- You, Y., Kanawade, V. P., de Gouw, J. A., et al.: Atmospheric amines and ammonia measured with a chemical ionization mass spectrometer (CIMS), Atmos. Chem. Phys., 14, 12181–12194, https://doi.org/10.5194/acp-14-12181-2014, 2014.
- Yoo, H., Lee, H., and Kim, Y. P.: Insights from single-particle analysis: submicron aerosol composition in Seoul during KORUS-AQ, Atmos. Chem. Phys., 24, 853–872, https://doi.org/10.5194/acp-24-853-2024, 2024.
- Zhang, Q., Jimenez, J. L., Canagaratna, M. R., et al.: Ubiquity and dominance of oxygenated species in organic aerosols in anthropogenically-influenced Northern Hemisphere midlatitudes, Geophys. Res. Lett., 34, L13801, https://doi.org/10.1029/2007GL029979, 2007.
- Ziemann, P. J. and Atkinson, R.: Kinetics, products, and mechanisms of secondary organic aerosol formation from gas-phase reactions of organic compounds, Chem. Soc. Rev., 41, 6582–6605, https://doi.org/10.1039/C2CS35122F, 2012.
- Zheng, J., Ma, Y., Chen, M., et al.: Measurement of atmospheric amines and ammonia using the high-resolution time-of-flight chemical ionization mass spectrometry, Atmos. Environ., 102, 249–259, https://doi.org/10.1016/j.atmosenv.2014.12.019, 2015.

---

## Author Comment (AC3)

**Response to Reviewer # 2**

- 2 We appreciate the anonymous reviewer for the thoughtful reviews and comments. We have carefully
- 3 considered the suggestions and revised the manuscript accordingly. The reviewer comments are in blue,
- 4 our comments are in black, and modifications to the manuscript are in red.
- 5 Kim et al. present a study of aerosol chemistry during wintertime in Seoul in 2019. Building upon their
- 6 earlier publication in 2017, the authors conducted a dedicated field campaign focusing on aerosol volatility
- 7 and obtained several intriguing results that are highly relevant to understanding and modeling aerosol
- 8 processes in this megacity. One particularly interesting finding is that highly oxidized organic aerosols were
- 9 shown to be highly volatile, providing observational evidence for autoxidation and fragmentation processes
- occurring in the particle phase. The dataset is well analyzed, and the manuscript is clearly written. The
- study fits well within the scope of ACP, and I consider it suitable for publication as a research article, rather
- than a measurement.
- We appreciate the thoughtful reviews and comments. We appreciate the recognition that our findings,
- particularly the observational evidence for autoxidation and fragmentation leading to highly oxidized but
- volatile organic aerosols, are highly relevant to understanding and modeling aerosol processes in the region.
- We have carefully considered the suggestions and revised the manuscript accordingly. The reviewer
- 17 comments are in blue, our comments are in black, and modifications to the manuscript are in red.
- 18 Section Introduction: The Introduction could be further strengthened by expanding the background on
- 19 aerosol volatility. Including more context on previous volatility-related studies would help frame the
- 20 contribution of this work.
- 21 Thank you for this helpful suggestion. To clarify the aims and frame our contribution, we expanded the
- 22 Introduction to explain why OA volatility is central to gas-particle partitioning and model performance,
- and summarize frameworks that couple volatility and oxidation. These changes/additions help motivate our
- 24 TD-AMS approach and the winter focus of this study. Now the relevant section reads:
- 25 "Volatility is a key parameter for characterizing organic aerosol (OA) properties, as it governs gas-to-
- 26 particle partitioning behavior and directly influences particle formation yields (Sinha et al., 2023). The
- 27 classification of OA species based on their volatility—from extremely low-volatility (ELVOC) to semi-
- volatile (SVOC) and intermediate-volatility (IVOC) compounds—is central to the conceptual framework
- 29 of secondary OA (SOA) formation and growth (Donahue et al., 2006). It also affects atmospheric lifetimes
- and human exposure by determining how long aerosols remain suspended in the atmosphere (Glasius and
- 31 Goldstein, 2016). Therefore, accurately capturing OA volatility is essential for improving predictions of
- 32 OA concentrations and their environmental and health impacts. However, chemical transport models often
- significantly underestimate OA mass compared to observations (Matsui et al., 2009; Jiang et al., 2012; Li
- 34 et al., 2017), largely due to incomplete precursor inventories and simplified treatment of processes affecting
- 35 OA volatility. For instance, aging—through oxidation reactions such as functionalization and
- fragmentation—can significantly alter volatility by changing OA chemical structure (Robinson et al., 2007;
- 37 Zhao et al., 2016). Early volatility studies primarily utilized thermal denuders (TD) coupled with various
- detection instruments to investigate the thermal properties of bulk OA (Huffman et al., 2008). The
- 39 subsequent coupling of TD with the Aerosol Mass Spectrometer allowed for component-resolved volatility

40 measurements, providing critical, quantitative insight into the properties of OA factors (e.g., SV-OOA vs.

41 LV-OOA) across different regions (Paciga et al., 2016; Cappa and Jimenez, 2010). These componentresolved volatility data are often used to constrain the Volatility Basis Set (VBS)—the current state-of-theart framework for modeling OA partitioning and evolution (Donahue et al., 2006). However, a limitation

in many field studies is that the TD-AMS thermogram data are rarely translated into quantitative VBS

distributions for individual OA factors, which limits their direct use in chemical transport models.

46 Furthermore, the volatility of OOA during extreme haze conditions, where the expected inverse correlation

between oxidation (O:C) and volatility can break down (Jimenez et al., 2009), remains poorly characterized,

particularly in East Asia's highly polluted winter environments. A recent study in Korea further highlighted

the importance of accounting for such processes when interpreting OA volatility under ambient conditions

(Kang et al., 2023). Given its central role in OA formation, reaction, and atmospheric persistence, volatility

analysis is critical for bridging the gap between measurements and model performance."

52 Section 2.1: This section currently begins with a citation to the authors' earlier study, which may lead

readers to assume that the dataset is the same. However, the actual description of the 2019 field study only

appears several sentences later. To improve clarity and logical flow, I suggest first presenting the details of

55 the current field campaign and then referring back to the earlier study for context.

We appreciate the suggestion. We rewrote the opening of section 2.1 to first describe the 2019 field

campaign and site, followed by a brief pointer to our earlier winter study for background on the same

58 location. We believe this improves the logical flow and avoids confusion with the 2017 dataset. Now the

59 relevant section reads;

60 "We conducted continuous real-time measurements in Seoul, South Korea, from 28 November to 28

61 December 2019. The sampling site was located in the northeastern part of the city (37.60° N, 127.05° E),

approximately 7 km from the city center, surrounded by major roadways and mixed commercial-residential

land use. Air samples were collected at an elevation of approximately 60 meters above sea level, on the

fifth floor of a building. A detailed site description has been reported previously for winter Seoul (Kim et

al., 2017). During this period, the average ambient temperature was  $1.76 \pm 4.3$  °C, and the average relative

humidity (RH) was 56.9 ± 17.5%, based on data from the Korea Meteorological Administration

(http://www.kma.go.kr)."

68

69

75

42

44

45

48

49 50

54

57

62 63

64

66

67

Line 146-150: This paragraph relies on information from the Supplementary Material, which makes it

awkward as an entry point into the main results. I suggest either removing it or integrating the content later

71 in the manuscript, once the main results are introduced.

72 Thank you for the helpful suggestion. We agree that starting a section by referencing time series data in the

73 Supplementary Material (Fig. S6) made the initial description awkward. We have reorganized the opening

74 of the Results section (section 3.1). The revised introductory paragraph now immediately establishes the

campaign's characteristics, providing the necessary meteorological and concentration context before

detailing the composition in Fig. 1. The revised section now reads:

- "We conducted continuous measurements from 28 November to 28 December 2019, characterizing a winter 77 period with a mean PM1 concentration of 27.8±15.3µgm-3. This concentration is characterized as moderate; 78
- 79 it closely matches historical winter PM1 means in Seoul (Kim et al., 2017) and implies an equivalent PM2.5
- concentration is about 34.8µgm-3 (using a Korea-specific PM1/PM2.5≈0.8 (Kwon et al., 2023)), which is 80
- near the national 24-h PM2.5 standard (35µgm-3) (AirKorea). The full co-evolution of PM1, gaseous 81
- 82 pollutants, and meteorological conditions is provided in Fig. S6, showing an average ambient temperature
- of 1.76±4.3°C and average relative humidity (RH) of 56.9±17.5% during the study. 83
- Figure 1 summarizes the overall non-refractory submicron aerosol (NR-PM1) composition and the 84
- identified OA factors.~" 85
- 86 Section 3.1.1: The identification of nitrogen-containing organic aerosols (NOA) could be better supported.
- 87 I encourage the authors to provide additional evidence, for instance through mass spectral comparison with
- previous studies, or by applying the NO/NO2 ratio approach to assess NOA, and then comparing the results 88
- 89 with PMF-based identification.
- 90 Thank you for the helpful suggestions. We agree that the identification of NOA must be clearly supported.
- Regarding the NO+/NO2+ ratio: This metric is a well-established diagnostic for assessing the thermal 91
- 92 decomposition (and thus functionality) of inorganic or organic nitrate (NO3-) in the AMS nitrate channel.
- 93 It is not applicable to reduced-nitrogen amines (R-NH2) that define NOA, which are detected in the organic
- spectrum as reduced CxHyN+ fragments (e.g., m/z30,44,58,86). Accordingly, we did not apply the 94
- 95 NO+/NO2+ method to the NOA factor. In order to strengthen the mass spectral evidence as requested, we
- have enhanced the discussion in 3.1.1 to clearly emphasize the spectral matching of our NOA factor against 96
- established literature reference spectra of amines (Fig. 3). Now the relevant section reads; 97
- 98 "The NOA factor exhibited the highest nitrogen-to-carbon (N:C) ratio (0.22) and the lowest oxygen-to-
- 99 carbon (O:C) ratio (0.19) among all POA factors (Fig. S2), indicating a chemically reduced, nitrogen-rich
- 100 composition. The factor represents semi-volatile, reduced nitrogen species that originate from primary
- urban combustion sources but whose observed mass in the particle phase is enhanced by rapid secondary 101
- 102 partitioning and salt formation (Ge et al., 2011; You et al., 2014). The NOA mass spectrum was dominated
- by amine-related fragments including m/z 30 (CH4N+), 44 (C2H6N+), 58 (C3H8N+), and 86 (C5H12N+) 103
- (Fig. 3a). The spectral signature of the factor is defined by the characteristic dominance of the m/z 44 104
- fragment, which typically serves as the primary marker for dimethylamine (DMA)-related species, closely 105
- 106 followed by m/z 58 (trimethylamine, TMA) and m/z 30 (methylamine, MA). This profile is in strong
- 107 agreement with NOA factors resolved via PMF in other polluted environments. For instance, the dominance
- of m/z 44 and m/z 30 aligns with amine factors reported in New York City (Sun et al., 2011) and Pasadena, 108
- California (Hayes et al., 2013). This DMA-dominated signature is also consistent with seasonal 109
- 110 characterization of organic nitrogen in Beijing (Xu et al., 2017) and Po Valley, Italy (Saarikoski et al.,
- 2012), reinforcing the common chemical signature of reduced organic nitrogen across diverse urban and 111
- 112 regional environments. Furthermore, the presence of non-negligible signals at m/z 86 supports
- the contribution of slightly larger alkylamines, a pattern that aligns well with established AMS laboratory 113
- 114 reference spectra for these reduced nitrogen compounds (Ge et al., 2011; Silva et al., 2008)"

- Line 187-188: Please clarify whether the identified NOA is of primary or secondary origin.
- 116 Thank you for requesting this clarification. The NOA factor exhibits characteristics of both primary
- emissions and rapid secondary processing, which is typical for reduced nitrogen species. We have clarified
- in the manuscript that the factor is best characterized as a semi-volatile component derived primarily from
- combustion emissions whose particle-phase concentration is enhanced by atmospheric processing.
- We base this conclusion on the following evidence.

121

122

123

124125

126

127128

129

130

131

- 1. Primary Origin (Precursors): The mass spectrum (Fig. 3) and elemental ratios (O:C=0.19, low oxidation) closely match low-molecular-weight alkylamines (e.g., DMA, TMA), which are primarily emitted during high-temperature combustion of nitrogen-rich fuels (Ge et al., 2011). The night-early morning peaks (Fig. 2) further link it to near-source urban combustion/heating activities.
- 2. Secondary Enhancement (Observed Mass): The observed high particle-phase concentration is strongly influenced by secondary processes:
  - o Partitioning: Amines are semi-volatile and highly basic. Their particle-phase retention (NOA mass) relies on partitioning facilitated by low temperature and reaction with acidic species to form low-volatility aminium salts (a secondary process).
  - o Meteorological Dependence: We observed a strong increase in NOA when relative humidity (RH) surpassed 60% (Fig. 2), suggesting that RH-driven partitioning and enhanced formation of aminium salts are critical for its detection (Milic et al., 2016).
- We have revised the text to clearly articulate this dual nature, emphasizing that while the precursor is primary, the particle mass is semi-secondary (governed by partitioning).
- 135 "The NOA factor exhibited the highest nitrogen-to-carbon (N:C) ratio (0.22) and the lowest oxygen-to-
- carbon (O:C) ratio (0.19) among all POA factors (Fig. S2), indicating a chemically reduced, nitrogen-rich
- 137 composition. The factor represents semi-volatile, reduced nitrogen species that originate from
- primary urban combustion sources but whose observed mass in the particle phase is enhanced by
- rapid secondary partitioning and salt formation (Ge et al., 2011; You et al., 2014).
- Line 212-213: The abbreviations for OOAs have already been introduced earlier.
- We removed repeated definitions of OOA and OA.
- Line 267: The abbreviation "OA" has also been defined earlier.
- We removed repeated definitions of OOA and OA.
- Line 268: ... observations at where?
- Thank you for pointing this out. We had not explicitly stated the location. The observations were made in
- **Seoul, Korea**. We have clarified the sentence accordingly.
- "This trend aligns with previously reported spring and fall observations in Seoul, Korea (Kang et al., 2022;
- 148 Jeon et al., 2023)."
- 149 Line 292: Section 3.3?

- 150 Corrected. Thanks.
- Line 302-303: Please add a few references to situate your results in the context of previous literature.
- We added references demonstrating that functionalization typically lowers volatility, while fragmentation
- increases it and can reduce SOA yields despite higher O:C; this supports our emphasis on pairing
- composition with direct volatility constraints. Now the section reads:
- "Generally, the oxygen-to-carbon (O:C) ratio of organic aerosols (OA) is inversely related to their volatility.
- As O:C increases through aging, the effective saturation concentration (C\*) typically decreases, resulting
- in lower volatility (Donahue et al., 2006; Jimenez et al., 2009). This common relationship arises because
- the addition of oxygen-containing functional groups (e.g., hydroxyl, carboxyl, carbonyl), which increases
- molecular weight and enhances intermolecular interactions such as hydrogen bonding, thereby reducing
- vapor pressure (Jimenez et al., 2009; Kroll and Seinfeld, 2008). Moreover, oxidative aging often leads to
- oligomerization or functionalization, promoting particle-phase retention and reducing the effective
- saturation concentration (C\*) (Donahue et al., 2011; Robinson et al., 2007). However, in this study, the
- most oxidized OA factor—MO-OOA, with a high O:C ratio of 1.15—exhibited unexpectedly high
- volatility. Its volatility distribution was skewed toward SVOCs and IVOCs (Fig. 5), and its rapid mass loss
- in MFR thermograms (Fig. S9) further indicated low thermal stability. This observation appears to
- 166 contradict the usual inverse O:C-volatility relationship; however, under winter haze conditions—with
- suppressed O3/low OH, particle-phase autoxidation and fragmentation can yield higher-O:C yet more
- volatile products, with enhanced condensation on abundant particle surface area (details below)."

**170 References**

169

171172

173

174175

176

177

178179

180

181

182

183

184

- Sinha, A., George, I., Holder, A., Preston, W., Hays, M., Grieshop, A. P., 2023. Development of volatility distributions for organic matter in biomass burning emissions. Environ. Sci. Adv. 3, 11–23. https://doi.org/10.1039/D2EA00080F
- Donahue, N. M., Robinson, A. L., Stanier, C. O., Pandis, S. N., 2006. Coupled partitioning, dilution, and chemical aging of semivolatile organics. Environ. Sci. Technol. 40, 2635–2643. https://doi.org/10.1021/es052297c
  - Glasius, M., Goldstein, A. H., 2016. Recent discoveries and future challenges in atmospheric organic chemistry. Environ. Sci. Technol. 50, 2754–2764. https://doi.org/10.1021/acs.est.5b05105
- Matsui, H., Koike, M., Takegawa, N., Kondo, Y., Griffin, R. J., Miyazaki, Y., Yokouchi, Y., Ohara, T., 2009. Secondary organic aerosol formation in urban air: Temporal variations and possible contributions from unidentified hydrocarbons. J. Geophys. Res. Atmos. 114, D02209. https://doi.org/10.1029/2008JD010164
- Jiang, F., Liu, Q., Huang, X., Wang, T., Zhuang, B., Xie, M., 2012. Regional modelling of secondary organic aerosol over China using WRF/Chem. J. Aerosol Sci. 53, 50–61. https://doi.org/10.1016/j.jaerosci.2011.09.003

- Li, J., Zhang, M., Wu, F., Sun, Y., Tang, G., 2017. Assessment of the impacts of aromatic VOC emissions and yields of SOA on SOA concentrations with the air quality model RAMS-CMAQ. Atmos. Environ. 158, 105–115. https://doi.org/10.1016/j.atmosenv.2017.03.035
- Robinson, A. L., Donahue, N. M., Shrivastava, M. K., Weitkamp, E. A., Sage, A. M., Grieshop, A. P., Lane, T. E., Pierce, J. R., Pandis, S. N., 2007. Rethinking organic aerosols: Semivolatile emissions and photochemical aging. Science 315, 1259–1262. https://doi.org/10.1126/science.1133061

- Zhao, B., Wang, S., Donahue, N. M., Jathar, S. H., Huang, X., Wu, W., ... & Hao, J. (2016). Quantifying the effect of organic aerosol aging and intermediate-volatility emissions on regional-scale aerosol pollution in China. Scientific Reports, 6, 28815. <a href="https://doi.org/10.1038/srep28815">https://doi.org/10.1038/srep28815</a>
- Huffman, J. A., Ziemann, P. J., Jayne, J. T., and Worsnop, D. R.: Development and characterization of a fast-stepping thermodenuder for chemically resolved aerosol volatility measurements, Aerosol Sci. Technol., 43, 1–15, <a href="https://doi.org/10.1080/02786820802104981">https://doi.org/10.1080/02786820802104981</a>, 2008.
- Huffman, J. A., Ziemann, P. J., Jayne, J. T., and Worsnop, D. R.: Development and characterization of a fast-stepping thermodenuder for chemically resolved aerosol volatility measurements, Aerosol Sci. Technol., 43, 1–15, <a href="https://doi.org/10.1080/02786820802104981">https://doi.org/10.1080/02786820802104981</a>, 2008.
- Paciga, A., Karnezi, E., Kostenidou, E., et al.: Volatility of organic aerosol and its components in the megacity of Paris, Atmos. Chem. Phys., 16, 2013–2031, <a href="https://doi.org/10.5194/acp-16-2013-2016">https://doi.org/10.5194/acp-16-2013-2016</a>, 2016.
- Donahue, N. M., Robinson, A. L., Stanier, C. O., Pandis, S. N., 2006. Coupled partitioning, dilution, and chemical aging of semivolatile organics. Environ. Sci. Technol. 40, 2635–2643. https://doi.org/10.1021/es052297c
- Jimenez, J. L., Canagaratna, M. R., Donahue, N. M., Prevot, A. S. H., Zhang, Q., Kroll, J. H., ... Worsnop, D. R., 2009. Evolution of organic aerosols in the atmosphere. Science 326, 1525–1529. https://doi.org/10.1126/science.1180353
- Kang, H. G., Kim, Y., Collier, S., Zhang, Q., Kim, H., 2022. Volatility of springtime ambient organic aerosol derived with thermodenuder aerosol mass spectrometry in Seoul, Korea. Environ. Pollut. 310, 119203. <a href="https://doi.org/10.1016/j.envpol.2022.119203">https://doi.org/10.1016/j.envpol.2022.119203</a>
- Kwon, S., Won, S. R., Lim, H. B., et al.: Relationship between PM1.0 and PM2.5 in urban and background areas of the Republic of Korea, Atmos. Pollut. Res., 14, 101858, https://doi.org/10.1016/j.apr.2023.101858, 2023.
- Kim, H., Zhang, Q., Bae, G.-N., et al.: Sources and atmospheric processing of winter aerosols in Seoul, Korea: insights from real-time measurements using a high-resolution aerosol mass spectrometer, Atmos. Chem. Phys., 17, 2009–2033, https://doi.org/10.5194/acp-17-2009-2017, 2017.
- Ge, X., Wexler, A. S., and Clegg, S. L.: Atmospheric amines Part II. Thermodynamic properties and gas-particle partitioning, Atmos. Chem. Phys., 11, 55–69, <a href="https://doi.org/10.5194/acp-11-55-2011">https://doi.org/10.5194/acp-11-55-2011</a>, 2011.
- P.L. Hayes, A.M. Ortega, M.J. Cubison, W.W. Hu, D.W. Toohey, J.H. Flynn, B.L. Lefer, N. Grossberg, S. Alvarez, B. Rappengl&ück, J.W. Taylor, J.D. Allan, J.S. Holloway, J.B. Gilman, W.C. Kuster, J.A. de Gouw, P. Massoli, X. Zhang, J. Liu, R.J. Weber, A.L. Corrigan, L.M. Russell, Y. Zhao, S.S. Cliff, G. Isaacman, D.R. Worton, N.M. Kreisberg, S.V. Hering, A.H. Goldstein, R. Thalman, E.M. Waxman, R. Volkamer, Y.H. Lin, J.D. Surratt, T.E. Kleindienst, J.H. Offenberg, K.D. Froyd, S. Dusanter, S. Griffith, P.S. Stevens, J. Brioude, W.M. Angevine, and J. L. Jimenez. Aerosol Composition and Sources in Los Angeles During the 2010 CalNex Campaign. *Journal of*

*Geophysical Research-Atmospheres*, 118, 9233-9257, May 2013. doi: 10.1002/jgrd.50530 Paper, 2013 and PDF. Figures (.zip) (August 27, 2013)

- Xu, W., Sun, Y., Wang, Q., et al.: Seasonal Characterization of Organic Nitrogen in Atmospheric Aerosols Using High Resolution Aerosol Mass Spectrometry in Beijing, China, ACS Earth Space Chem., 1, 649–658, <a href="https://doi.org/10.1021/acsearthspacechem.7b00106">https://doi.org/10.1021/acsearthspacechem.7b00106</a>, 2017.
  - Sun, Y. L., Zhang, Q., Schwab, J. J., et al.: Characterization of the sources and processes of organic and inorganic aerosols in New York city with a high-resolution time-of-flight aerosol mass apectrometer, Atmos. Chem. Phys., 11, 1581–1602, <a href="https://doi.org/10.5194/acp-11-1581-2011">https://doi.org/10.5194/acp-11-1581-2011</a>, 2011.
  - Xu, W., Sun, Y., Wang, Q., et al.: Seasonal Characterization of Organic Nitrogen in Atmospheric Aerosols Using High Resolution Aerosol Mass Spectrometry in Beijing, China, ACS Earth Space Chem., 1, 649–658, <a href="https://doi.org/10.1021/acsearthspacechem.7b00106">https://doi.org/10.1021/acsearthspacechem.7b00106</a>, 2017.
  - Saarikoski, S., Carbone, S., Decesari, S., et al.: Chemical characterization of springtime submicrometer aerosol in Po Valley, Italy, Atmos. Chem. Phys., 12, 8401–8421, https://doi.org/10.5194/acp-12-8401-2012, 2012.
  - Silva, P. J., Erupe, M. E., Price, D., et al.: Trimethylamine as Precursor to Secondary Organic Aerosol Formation via Nitrate Radical Reaction in the Atmosphere, Environ. Sci. Technol., 42, 4689–4696, https://doi.org/10.1021/es703016v, 2008.

---

## Author Response (AR2)

We appreciate the anonymous reviewer for the thoughtful reviews and comments. We have carefully considered the suggestions and revised the manuscript accordingly. The reviewer comments are in blue, our comments are in black, and modifications to the manuscript are in red.

General comments:

Kim et al. have substantially improved the manuscript, and the dataset is valuable. However, several structural, interpretive, and technical issues remain that should be addressed before acceptance in Atmospheric Chemistry and Physics. Below I list Major (structural/scientific) items followed by Specific & Technical (line- or figure-level) points.

We sincerely thank the Editor for the thoughtful and constructive feedback on our manuscript. We also appreciate the reviewers' supportive comments recommending acceptance. In response to the Editor's request for structural and narrative improvements, we have carefully revised the manuscript to enhance clarity, flow, and conciseness while maintaining the scientific depth.

Major revisions include:
(1) Complete reorganization of Section 3.1 to follow Overview → SOA → POA → NOA;
(2) Clearer explanation of how the NOA factor was identified and why it emerged in this study;
(3) Reordering and refinement of Section 3.3.1 with a focus on results and evidence-based interpretation; and
(4) Rewriting of the Conclusion to emphasize the two principal findings (NOA and volatile MO-OOA).

We also performed comprehensive copy-editing and addressed all figure-level and line-level comments. Details are provided below.

Major comments

1. Restructure Section 3.1 (clarity and reader flow).

Because resolving NOA is central to the manuscript and many readers may be unfamiliar with this factor, I recommend reorganizing Section 3.1 as: Overview → SOA → POA → NOA. This order (from broadly known to newly resolved) will help readers understand the factors that are consistent with previous studies before introducing the novel NOA factor. As NOA is closely related to POA, discuss NOA immediately after POA.

We appreciate this valuable suggestion. Section 3.1 has been completely reorganized to follow the requested sequence, improving the narrative flow and reader comprehension. The section now begins with a short overview of total $PM_1$ composition, followed by SOA, POA, and finally NOA.

**Revised text (excerpt):**

"Among the organic aerosols, six OA factors were identified during the winter of 2019: hydrocarbon-like OA (HOA; 14%; O:C = 0.13), cooking-related OA (COA; 21%; O:C = 0.18), nitrogen-enriched OA (NOA; 2%; O:C = 0.22), biomass burning OA (BBOA; 13%; O:C = 0.25), and two types of secondary organic aerosols—less-oxidized oxygenated OA (LO-OOA; 30%; O:C = 0.68) and more-oxidized oxygenated OA (MO-OOA; 20%; O:C = 1.15) (Fig. 1e and Fig. S2). These compositions are consistent with previous wintertime observations in Kim et al. (2017), with the exception of newly resolved NOA source. In the following sections, we describe each OA factor in the order of secondary OA (SOA), primary OA (POA)

and finally introduce NOA, which—while related to combustion POA—emerged as a distinct, nitrogen-rich factor under the winter conditions of this study~."

2. Clarify how NOA was identified and why it appears now.

In Section 3.1.1, first explain the diagnostics that establish the NOA factor (spectral markers, HR signatures, diurnal behavior, correlations). After that, discuss why NOA was resolved during this campaign but not in prior campaigns in Seoul (e.g., instrumental mode/resolution differences, episodic source enhancement, seasonality, PMF settings). And lastly, compare NOA characteristics with observations in other cities to contextualize uniqueness vs. commonality.

We appreciate this thoughtful and constructive suggestion. The original manuscript included several elements of the NOA diagnosis and inter-campaign comparison, but these were distributed across sections and not presented in a clear order. In the revised version, Section 3.1.1 has been **substantially reorganized** to follow the structure recommended by the reviewer:

**Revised text (excerpt):**

"The NOA factor was identified based on distinct spectral markers, high-resolution chemical signatures, and diurnal behavior. Chemically, this factor exhibited the highest nitrogen-to-carbon (N:C) ratio (0.22) and the lowest oxygen-to-carbon (O:C) ratio (0.19) among all OA factors (Fig. S2), indicating a chemically reduced, nitrogen-rich composition. The high-resolution mass spectrum was dominated by amine-related fragments, including m/z 30 ($CH_4N^+$), 44 ($C_2H_6N^+$), 58 ($C_3H_8N^+$), and 86 ($C_5H_{12}N^+$) (Fig. 3a). Specifically, the spectral signature is defined by the characteristic dominance of the *m/z* 44 fragment—typically the primary marker for dimethylamine (DMA)-related species—closely followed by *m/z* 58 (trimethylamine, TMA) and *m/z* 30 (methylamine, MA). The presence of non-negligible signals at *m/z* 58 and *m/z* 86 further supports the contribution of slightly larger alkylamines (Ge et al., 2011; Silva et al., 2008).

In terms of temporal behavior, NOA displayed a diurnal pattern similar to that of BBOA, with both peaking at night and in the early morning (Fig. 2a). Strong correlations with amine fragments $CH_4N^+$ (r = 0.95) and $C_2H_6N^+$ (r = 0.91) (Fig. 2) confirm that this factor represents reduced nitrogen compounds. While amines can originate from industrial sources or solvent use (e.g., TEA, DEA) (Tiszenkel et al., 2024; Zheng et al., 2015), the dominance of small alkylamines and the covariation with combustion markers suggest a primarily combustion-linked influence, likely associated with biomass burning or residential heating (Ge et al., 2011; You et al., 2014). However, the time series of NOA and BBOA were not perfectly correlated (Fig. 2 and S7), likely because NOA episodes were preferentially enhanced during stagnant haze periods (Fig. 1), whereas BBOA emissions followed a more regular daily emission pattern.

Although nitrogen-containing species have been identified in Seoul via filter-based analysis (Baek et al., 2022), PMF-based resolution of NOA in real-time has not been reported in prior wintertime studies. The successful isolation of NOA in this study is likely driven by the synergistic effect of stagnation and favorable thermodynamic conditions, which significantly elevated particle-phase concentrations. The detection of NOA is often limited by its low mass concentration and high volatility; however, in this campaign, persistent stagnation caused pollutants to accumulate, frequently elevating NOA concentrations above 1 μg m$^{-3}$ during haze events (Fig.2). Simultaneously, the consistently cold and humid winter conditions (T = –0.24 °C; RH ≈ 57%) favored the gas-to-particle partitioning of semi-volatile amines (Fig. S2), ensuring that this accumulated mass remained in the particle phase. This combination of physical accumulation and thermodynamic partitioning improved the signal-to-noise ratio, allowing the HR-ToF-AMS to mathematically distinguish the NOA factor from the bulk organic aerosol—a separation that relies

on the instrument's ability to resolve isobaric nitrogen-containing ions from hydrocarbon fragments once concentrations are sufficiently high.

The NOA factor contributed approximately 2% of the total organic aerosol (OA) mass, a fraction comparable to observations in Guangzhou (3%; Chen et al., 2021), Pasadena (5%; Hayes et al., 2013), and New York (5.8%; Sun et al., 2011). The spectral profile identified in Seoul is in strong agreement with NOA factors resolved in these diverse environments. For instance, the dominance of $m/z$ 44 and $m/z$ 30 aligns with amine factors reported in New York City (Sun et al., 2011) and Pasadena (Hayes et al., 2013). This DMA-dominated signature is also consistent with the seasonal characterization of organic nitrogen in Beijing (Xu et al., 2017) and Po Valley, Italy (Saarikoski et al., 2012). Taken together, these results confirm that the NOA resolved in Seoul shares a common chemical signature with reduced organic nitrogen observed globally, representing semi-volatile species that originate from primary urban combustion but are enhanced in the particle phase by rapid secondary partitioning."

3. Reorder Section 3.3.1 and avoid overreaching language.

Present results first, then discussion. Limit speculative material and ensure that mechanistic claims are supported by measurements or cited modeling. The current Section 3.3.1 contains speculative text that is not directly supported by the manuscript's measurements. Retain the essential discussion (the paragraph that addresses reasons for higher volatility of MO-OOA) but shorten and focus the first paragraph; place tentative hypotheses in the Discussion with clear labeling as speculation.

I recommend avoiding the word "Mechanism" in the subsection title unless direct mechanistic evidence is presented. A more suitable and descriptive title would be something like "High-volatility nature of MO-OOA in Seoul wintertime."

We have restructured Section 3.3.1 to follow a clear results → interpretation → speculation (labeled) format. The subsection title was changed from Mechanism of MO-OOA volatility to High-volatility nature of MO-OOA in Seoul wintertime. The revised section is now concise and evidence-driven, with speculative content explicitly labeled.

**Revised text:**

"MO-OOA exhibited high O:C ratios and high apparent volatility, characteristics that were further amplified during haze episodes—periods marked by reduced ozone levels, low solar radiation, and elevated aerosol mass concentrations (Fig. 6 and Fig. S6, yellow shading). Spectrally, MO-OOA was defined by a consistently high $f_{44}$ ($CO_2^+$) signal and a comparatively stable $f_{43}$ ($C_2H_3O^+$) signal relative to LO-OOA (Fig. S8b). Notably, when MO-OOA concentrations intensified during haze, only $f_{44}$ was significantly enhanced, while $f_{43}$ remained nearly unchanged (Fig. 6). This trend is corroborated by the haze–non-haze comparison (Fig. S12), where haze periods (including high MO-OOA intervals) showed elevated contributions from oxygenated fragments ($m/z$ 28, 29, 44) and higher O:C ratios. In contrast, non-haze periods were characterized by larger fractional contributions from hydrocarbon-like fragments ($m/z$ 41, 43, 55, 57). The observed temporal pattern—elevated $f_{44}$ without corresponding changes in $f_{43}$—is a typical signature of highly oxidized and fragmented organic aerosol, suggesting that aging was dominated by fragmentation rather than functionalization (Kroll et al., 2009). These spectral patterns collectively indicate that MO-OOA is highly oxidized yet remains relatively volatile compared to LO-OOA.

The elevated volatility of MO-OOA despite its high O:C (~1.15) indicates that oxidation under these haze conditions did not follow the classical multi-generational OH-driven aging pathway, which typically increases molecular mass and reduces volatility. Instead, the data align with fragmentation-dominated aging, where highly oxygenated but lower-molecular-weight compounds (e.g., small acids or diacids) are formed. Prior field and laboratory studies using online AMS/FIGAERO-CIMS and EESI-TOF have similarly reported high-O:C yet volatile product distributions characterized by high $f_{44}$ and stable $f_{43}$ (Kroll et al., 2009; Ng et al., 2010; Chhabra et al., 2011; Lambe et al., 2012; Lopez-Hilfiker et al., 2016; D'Ambro et al., 2017).

While direct mechanistic measurements were not available in this study, we hypothesize that the formation of this volatile, high-O:C component may be driven by specific low-light oxidation pathways consistent with the observed environmental conditions. The suppressed ozone levels during haze likely indicate a low-OH oxidation regime (Fig. 6). Under such conditions, radical chemistry involving $NO_3$ (which is longer-lived in low light) or particle-phase autoxidation could preferentially produce highly oxygenated but relatively small organic fragments (Ehn et al., 2014; Zhao et al., 2023). Although haze suppresses photolysis, HONO concentrations—maintained via heterogeneous conversion or surface emissions—could still provide a non-negligible source of OH (Gil et al., 2021; Kim et al., 2024; Slater et al., 2020). Furthermore, the high aerosol mass loadings during haze ($C_{oa}$) provide abundant surface area for absorptive partitioning (Pankow, 1994; Donahue et al., 2006). This increased partitioning mass allows even relatively volatile, oxidized compounds to condense into the particle phase, contributing to the high apparent volatility and oxidation state observed (Jimenez et al., 2009; Ng et al., 2016). Consequently, these results underscore the need for SOA models to incorporate fragmentation-dominated pathways to accurately represent wintertime haze evolution."

4. Revise the Conclusion to emphasize NOA and the volatile MO-OOA observation.

The first paragraph of the Conclusion currently omits a balanced discussion of NOA's mixed primary/secondary nature and overemphasizes biomass burning. Combine the first two paragraphs and use that space to emphasize the manuscript's principal takeaways: (a) reliable identification of a NOA factor and its mixed origin, and (b) the observation that MO-OOA in Seoul exhibits unexpectedly high volatility. Frame outstanding uncertainties and recommended follow-up analyses/measurements as clear next steps.

We agree and have rewritten the Conclusion to highlight these two main contributions and add concise statements on implications and next steps.

**Revised text:**

"This study provides a comprehensive characterization of wintertime submicron aerosols ($PM_1$) in Seoul, integrating chemical composition, volatility measurements, and source apportionment to reveal critical insights into urban OA evolution. The two most significant findings are the robust real-time identification of a nitrogen-containing organic aerosol (NOA) factor and the observation of unexpected volatility behavior in highly oxidized OA. The NOA factor, spectrally dominated by low-molecular-weight alkylamine fragments, was successfully resolved primarily due to the accumulation of pollutants during wintertime stagnation, which sufficiently enhanced the spectral signals of these semi-volatile species for identification. Its temporal and chemical characteristics point to a mixed primary/secondary origin: driven by direct combustion emissions (e.g., residential heating) but significantly enhanced by the rapid gas-to-particle partitioning of semi-volatile amines under cold, humid conditions. Concurrently, the volatility analysis revealed a striking decoupling between oxidation state and volatility for the More-Oxidized Oxygenated

OA (MO-OOA). Despite its high O:C ratio (~1.15), MO-OOA exhibited elevated volatility, a deviation from classical aging models that typically associate high oxidation with low volatility. This behavior is attributed to the specific conditions of winter haze—reduced photolysis and high aerosol mass loadings—which favor fragmentation-dominated aging pathways and the absorptive partitioning of volatile oxygenated products.

These results revise our understanding of wintertime aerosol dynamics and underscore the limitations of current models in representing reduced-nitrogen species and non-canonical oxidation pathways. To address the remaining uncertainties, future research should prioritize evaluating the seasonal variability of NOA to better disentangle the influence of meteorological drivers from specific emission sources. Concurrently, there is a critical need to directly probe radical oxidation mechanisms, such as $RO_2$ autoxidation and $NO_3$ chemistry, particularly under haze conditions. Integrating these field inquiries with laboratory studies and advanced molecular-level measurements (e.g., FIGAERO-CIMS, EESI-TOF) will be essential for constraining the formation, lifetime, and climate impacts of these complex organic aerosol components in polluted megacities."

Specific & technical comments (line- and figure-level)

Manuscript-wide editorial/formatting

There are numerous minor formatting and grammar issues (missing spaces after punctuation, double punctuation, inconsistent font sizes, inconsistent spacing around "m/z", missing italics for m/z and f##). Please perform a careful copy-edit before the next submission.

We performed a detailed copy-edit to correct punctuation spacing, double punctuation, inconsistent fonts, "m/z" and "f##" formatting, and italicization.

Figures/visualization

Line 186 / Fig. S8: Cluster 1 is not well visualized in Fig. S8. Provide a zoomed-in version (or higher-resolution panel) so the cluster interpretation is clear.

We agree that the resolution of Cluster 1 in the original figure was insufficient. To address this, we have modified Figure S8 to include a zoomed-in inset that clearly displays the trajectory details for this cluster. The revised figure is shown below.

[Figure]

Line 409 / f44 and f43: The high and stable f44 and f43 results are important and should be shown in a main-text figure rather than only in the supplement. If retained in the supplement, clearly reference them in the main text and ensure they are visible in the linked supplement figure.

We appreciate the reviewer's suggestion. We agree that the temporal stability of $f_{44}$ and $f_{43}$ is central to understanding the volatility of MO-OOA. Accordingly, we have moved the key time-series data from the supplement to the main text (Figure 6) to ensure they are prominent and directly accessible alongside the relevant discussion.

PMF/factor robustness

• Line 201: Because the NOA mass fraction is low, please report PMF reproducibility metrics (bootstrap). If bootstrap results exist, include them and summarize whether NOA is reproducibly resolved. If not performed, run bootstrap tests to quantify factor stability.

Thank you for your suggestions. We performed bootstrapping to ensure that the PMF solution is robust and stable. The procedure of bootstrap was updated into method section. Relevant figures are updated at supplementary Table S2 and Figure S13 as follows:

"To ensure the robustness of the 6-factor solution, we calculated uncertainties for each PMF factor using the bootstrap method (100 iterations) with the PET toolkit (v2.05) (EPA, 2014; Xu et al., 2018; Srivastava et al., 2021). This method generates a time series distribution for each factor, providing an average

concentration and standard deviation; the uncertainty is defined as the standard deviation divided by the average concentration. As shown in Table S2, the 5-factor solution exhibited the lowest average uncertainty (5.10%). While mathematically stable, this low uncertainty is typical of under-resolved solutions where distinct sources are merged. In the 6-factor solution, the average uncertainty increased slightly to 6.06%, with individual factors ranging from 4.26% (MO-OOA) to 9.36% (BBOA). Despite this marginal increase, all factors in the 6-factor solution remained well within the acceptable range (<10%), confirming that the separation of the additional source did not compromise the solution's statistical stability. In contrast, the 7-factor solution showed signs of instability, with the average uncertainty rising to 7.79% and specific factors exceeding 10% (e.g., Factor 3 at 13.32% and Factor 5 at 11.94%). This degradation suggests the splitting of a factor into non-robust artifacts. Therefore, the 6-factor solution was selected as the optimal choice, offering the best balance of chemical resolution and statistical robustness. The average concentration and $1\sigma$ variability for the chosen 6-factor solution are presented in Figure S13"

**Table S2.** Uncertainty in factor concentration for the 5 to 7-factor solution from 100 iterations bootstrap.

| | Factor 1 | Factor 2 | Factor 3 | Factor 4 | Factor 5 | Factor 6 | Factor 7 |
|---|---|---|---|---|---|---|---|
| 5-factor solution | 4.73% | 6.46% | 5.95% | 5.78% | 2.59%- | - | - |
| 6-factor solution | 4.26% (MO-OOA) | 5.23% (LO-OOA) | 9.36% (BBOA) | 6.48% (NOA) | 5.24% (COA) | 5.80% (HOA) | - |
| 7-factor solution | 5.45% | 4.97% | 13.32% | 7.09% | 11.94% | 4.85% | 6.90% |

[Figure]

**Figure S13.** Bootstrapping analysis of the 6-factor solution (average factor with 1σ variation for each point)

Repetition/consolidation

• Lines 208–212 vs. 229–250: These passages appear repetitive (NOA relevance to combustion). Combine and streamline these discussions to avoid redundancy. Place the definitive explanation and supporting evidence in one location.

The repeated passages have been consolidated into a single, concise explanation of NOA's link to combustion in Section 3.1.3.

Clarifications/wording

Line 257: Clarify "highly oxidized chemical composition." Do you mean "oxidized relative to POA" or "oxidized relative to other OA factors"? Be explicit.

We thank the reviewer for this helpful comment. We have clarified the intended comparison by explicitly stating the reference frame of oxidation. In the revised text, we now specify that the high O:C ratios of MO-OOA (1.15) and LO-OOA (0.68) indicate oxidation levels that are elevated relative to the primary OA

factors (HOA, COA, BBOA) and that MO-OOA is substantially more oxidized than LO-OOA. The revised sentence reads:

"The oxygen-to-carbon (O:C) ratios for MO-OOA and LO-OOA were 1.15 and 0.68, respectively, indicating that both factors are highly oxidized relative to the primary OA factors (HOA, COA, BBOA) and that MO-OOA is substantially more oxidized than LO-OOA."

• Line 264: The statement that "LO-OOA is mostly related to primary sources, but not secondary" is ambiguous. Also, wouldn't LO-OOA vs. MO-OOA be more related to "less aged" versus "more aged"?

We thank the reviewer for pointing out this ambiguity and for the insightful suggestion regarding the "aged" distinction. We agree that the terminology "primary vs. secondary" was confusing in this context.

We have revised the manuscript to adopt the reviewer's suggested framework, characterizing LO-OOA and MO-OOA based on their degree of atmospheric processing ("Less Aged" vs. "More Aged") rather than a strict source dichotomy. Our intention is to clarify that while MO-OOA represents highly processed regional aerosol, LO-OOA represents a less aged fraction (fresh SOA) that has not yet undergone the extensive oxidation required to correlate strongly with secondary inorganic species.

"In contrast, LO-OOA exhibited only modest correlations with sulfate, nitrate, and ammonium ($r = 0.50$, 0.51, and 0.42, respectively). This weaker coupling indicates that LO-OOA represents a less aged oxygenated OA component (fresh SOA), distinguishable from the more aged, highly processed MO-OOA which tracks closely with secondary inorganic species. Regarding potential primary influence, LO-OOA does not exhibit a pronounced $m/z$ 60 (levoglucosan) signal (Fig. S2). While the levoglucosan marker ($f_{60}$) is known to diminish with atmospheric aging and can become weak or undetectable downwind (Hennigan et al., 2010; Cubison et al., 2011), the absence of a distinct peak combined with the separation from inorganic salts suggests that LO-OOA is best characterized as freshly formed secondary organic aerosol likely originating from the rapid oxidation of local anthropogenic precursors."

• Line 280: When defining COA, report the m/z 55/57 ratio used as the key tracer for COA identification and show those values or rationale in the text or table.

We appreciate this suggestion. In the revised manuscript, we now explicitly report the $m/z$ 55/57 ratio used as a diagnostic for the COA factor and relate it to previous AMS studies to strengthen the identification.

"COA factor showed an enhanced signal at $m/z$ 55 relative to $m/z$ 57, with a 55/57 ratio of 3.11, substantially larger than that of HOA (1.10). This elevated ratio is consistent with previously reported AMS COA spectra in urban environments (e.g., Allan et al., 2010; Mohr et al., 2012; Sun et al., 2011), supporting our factor assignment."

Line 308: When you state "Nearly complete evaporation occurred by 200 °C (~2% remaining)", indicate explicitly which species you refer to (e.g., nitrate, organics) and include the figure/table reference.

We thank the reviewer for this helpful clarification request. We have revised the text to explicitly state that the ~2% remaining mass refers to the nitrate thermogram and have added the figure reference. Furthermore, we have expanded this section to discuss the $T_{50}$ shift relative to pure ammonium nitrate, clarifying that the residual fraction and the elevated volatilization temperature likely reflect contributions from low-volatility nitrate-containing species (e.g., organonitrates) rather than pure ammonium nitrate.

"Nitrate showed the steepest decline with increasing temperature, with a $T_{50}$ of ~67 °C—substantially higher than that of pure ammonium nitrate (~37 °C; Huffman et al., 2009). At 200 °C, ~2% of the initial nitrate signal remained (Fig. 4). Since pure ammonium nitrate fully evaporates well below this temperature (Huffman et al., 2009), this small residual fraction likely represents the least volatile portion of organic nitrates."

Lines 311–314: Statements in this range make inferences not directly shown by measurements. Add appropriate references or align the text with the data; avoid speculative assertions without citation.

We thank the reviewer for this helpful comment. We have revised the sulfate thermogram description to remove unsupported inferences and have added appropriate references documenting AMS limitations and sulfate volatility behavior. Specifically, we now clarify that the slope change near 140 °C is consistent with ammonium-sulfate phase transitions or mixed organic–inorganic sulfate interactions reported in previous TD-AMS studies, while explicitly acknowledging that metallic sulfates are not efficiently detected by the AMS.

"Sulfate exhibited the highest thermal stability among the measured species. The thermogram showed a relatively stable mass fraction (MFR > 0.8) up to ~130 °C, followed by a sharp decline at temperatures above 140 °C (Fig. 4). This profile is consistent with the typical volatilization behavior of ammonium sulfate in TD-AMS, which requires higher temperatures to evaporate compared to nitrate or organics (Huffman et al., 2009). At 200 °C, approximately 25% of the sulfate mass remained. This residual suggests the presence of a sulfate fraction with lower volatility than pure ammonium sulfate, likely associated with organosulfates or low-volatility mixtures, whereas refractory metal sulfates are not efficiently detected by the AMS (Canagaratna et al., 2007)."

• Line 318: Specify which chloride species you mean (e.g., ammonium chloride vs. metal chlorides) and provide literature references for volatility comparisons.

Thank you for this important point. We agree that the chloride discussion should be more explicit. We have clarified that the chloride measured by the AMS is expected to be dominated by volatile inorganic chloride, primarily ammonium chloride ($NH_4Cl$), which evaporates efficiently in thermodenuder systems. We have also clarified that metal chlorides (e.g., NaCl, KCl) are far less volatile and are poorly detected by AMS; therefore, they are unlikely to contribute significantly to the observed thermogram.

"Chloride volatility was broadly consistent with prior AMS studies, with $T_{50}$ values comparable across seasons (e.g., Xu et al., 2016; Jeon et al., 2023). The near-complete evaporation observed in winter (~4% residual at 200 °C, Fig. 4) indicates that the chloride measured here was dominated by volatile inorganic chloride, specifically ammonium chloride ($NH_4Cl$), which fully evaporates at relatively low temperatures (Huffman et al., 2009). By contrast, metal chlorides (e.g., NaCl, KCl) are refractory and far less volatile; they are also poorly detected by the AMS (Canagaratna et al., 2007). The lower residual in winter compared to fall (~10%) therefore suggests that wintertime chloride consisted almost exclusively of pure ammonium

chloride, whereas the fall samples may have contained a minor fraction of less volatile or refractory chloride species."

Lines 319–321: The current wording is unclear. If you intend to say that organics display a continuous decrease in mass fraction with TD temperature, indicating a range of volatilities, whereas some inorganic species show abrupt loss at specific temperatures, please rewrite explicitly and include references if appropriate.

We thank the reviewer for this helpful clarification request. We agree that the original wording was ambiguous. In the revised manuscript, we explicitly state that the organic thermogram exhibits a smooth and continuous decrease in mass fraction with increasing temperature, reflecting a broad range of volatilities. We now contrast this behavior with the more abrupt, species-specific evaporation transitions typical of inorganic components (e.g., nitrate, ammonium chloride) and have added relevant references.

"Organics exhibited moderate volatility ($T_{50}$ ~120 °C), and their thermogram showed a gradual, continuous decrease in mass fraction with increasing TD temperature. This smooth profile reflects the presence of a broad distribution of organic compounds spanning SVOC to LVOC ranges, in contrast to inorganic species such as nitrate or ammonium chloride, which often show more abrupt losses at characteristic temperatures (Huffman et al., 2009; Xu et al., 2016). This behavior is consistent with previous TD-AMS observations in Seoul during spring and fall (Kang et al., 2022; Jeon et al., 2023)."

Line 334: Oxidation is typically more active during daytime photochemistry. The current statement that tries to explain the higher O:C of HOA is not convincing enough.

We thank the reviewer for this helpful comment. We agree that the previous explanation was imprecise, as oxidation is generally driven by daytime photochemistry rather than nighttime conditions. Our intention was not to imply that these primary factors undergo enhanced atmospheric oxidation at night, but rather to highlight that their O:C ratios are determined by their primary source signatures. BBOA and NOA tend to exhibit slightly higher O:C ratios than HOA because they are emitted with more oxygenated functional groups (e.g., from biomass pyrolysis or amine structures), not because of secondary aging.

To avoid confusion, we have revised the text to remove the misleading reference to photochemistry and provide a clearer, emission-based explanation for the observed O:C differences.

"Among the primary OA (POA) sources, hydrocarbon-like OA (HOA) exhibited the highest volatility, with mass predominantly distributed in the SVOC and IVOC ranges, consistent with its chemically reduced nature (O:C = 0.13) and direct combustion origin. Mass fraction remaining (MFR) results (Fig. S9) further support this, showing rapid mass loss at lower temperatures. Biomass burning OA (BBOA) and nitrogen-containing OA (NOA) also showed high volatility, peaking in the SVOC–IVOC range (log $C^* = 1$–3), but displayed slightly higher O:C ratios (0.25 and 0.19, respectively). This modest enhancement in O:C reflects their source composition—biomass combustion produces partially oxygenated organic species (e.g., levoglucosan, phenols), and NOA contains nitrogen-bearing functional groups—rather than enhanced atmospheric oxidation."

Line 337: COA appears to be less oxidized than BBOA but also less volatile — please comment on this apparent discrepancy and add discussion.

We thank the reviewer for this insightful comment. We agree that COA displays a distinct volatility-oxidation relationship compared to other primary factors. We have revised the text to clarify that this behavior is consistent with previous TD-AMS studies, which attribute the lower volatility of COA to its specific molecular composition.

"Cooking-related OA (COA) showed a more moderate volatility profile, with mass more evenly distributed across the LVOC and SVOC bins. This behavior differs from that of BBOA, which is slightly more oxidized yet more volatile. This apparent decoupling between oxidation state and volatility is a characteristic feature of COA reported in previous volatility studies (Paciga et al., 2016; Kang et al., 2022). These studies attribute the lower volatility of COA to its abundance of high-molecular-weight fatty acids (e.g., oleic, palmitic, and stearic acids) and glycerides (Mohr et al., 2009; He et al., 2010). Unlike the smaller, fragmented molecules typical of biomass burning, these lipid-like compounds possess high molar masses that suppress volatility, even though their long alkyl chains result in low O:C ratios."

Lines 353–357: These sentences are repetitive. Condense to improve readability.

We thank the reviewer for this suggestion. We have condensed the text to avoid redundancy while retaining the physical explanation for the observed relationship.

"This relationship arises because oxidative functionalization introduces polar groups (e.g., hydroxyl, carboxyl) that increase molecular weight and enhance intermolecular hydrogen bonding, thereby reducing the effective saturation concentration ($C^*$) and promoting particle-phase retention (Jimenez et al., 2009; Kroll and Seinfeld, 2008; Donahue et al., 2011)."

Line 368: Clarify the phrase "strong overall OA volatility" — do you mean "higher volatility" or "greater mass loss upon heating"? Use precise language.

We appreciate the reviewer's comment. The original wording repeated similar ideas about high OA volatility across different cities and seasons, making the sentence longer than necessary. We have condensed this part to present the comparison more efficiently while retaining the essential context from previous TD-AMS studies.

"prior TD-AMS studies in Mexico City, Los Angeles, Beijing, and Shenzhen have all reported substantial SVOC–IVOC contributions during polluted periods, indicating that high OA volatility is a common feature of urban environments across seasons (Cappa and Jimenez, 2010; Xu et al., 2019; Cao et al., 2018)."

Other

• Line 418–421: Material here reads like a summary and is more appropriate for the Conclusion. Consider moving it there.

We thank the reviewer for pointing out that the material in Lines 418–421 previously read as summary-type content. In the revised manuscript, this text has been removed from Section 3.3.1 and incorporated into the Conclusion, consistent with the reviewer's recommendation. The final version of Section 3.3.1 now contains only measurement-based results and focused interpretation, without summary elements.

---

## Author Response (AR3)

**Response to the comments**

We sincerely thank the anonymous reviewer for the thoughtful review and constructive comments. We have carefully considered all suggestions and revised the manuscript accordingly. The reviewer comments are in blue, our comments are in black, and modifications to the manuscript are in red.

General comments:

As stated by the reviewer, the authors have revised the manuscript carefully in response to previous comments, and the paper has improved substantially. However, a few minor technical and editorial corrections are needed for improved clarity. Additional private note (visible to authors and reviewers only):Please address the minor comments provided by the reviewers.

We sincerely thank the Editor for the thoughtful and constructive feedback on our manuscript. We also appreciate the reviewer's positive recommendation for acceptance. In response to the request for minor revisions, we have corrected the manuscript as detailed below.

Minor Comments

Line 218: The authors may consider explicitly including NOA at this point and introducing a dedicated subsection (e.g., Section 3.1.2.1). This may improve clarity, but I leave the final decision to the authors.

Thank you for this helpful suggestion. We have restructured the discussion by introducing a dedicated subsection for NOA (Section 3.1.2.1) to improve clarity and organization.

Lines 243–245: Please point out which figure represents the trajectory that you mentioned here.

We agree that this was missing. We have now explicitly cited Fig. S8 in the revised text to refer to the relevant air mass back-trajectories.

Lines 301–305: This paragraph appears closely related to the discussion in the first paragraph of this section (Lines 247–261), where comparisons with other urban sites are already made. I suggest combining these paragraphs to improve conciseness and flow.

Thank you for the suggestion. Following the reviewer's advice, we have combined these two paragraphs to provide a more cohesive discussion of the chemical characterization, mass contribution, and regional comparisons of the NOA factor. The integrated section (now Section 3.1.2.1) reads:

A distinct nitrogen-containing organic aerosol (NOA) factor was resolved in this study, whereas earlier wintertime AMS–PMF analyses in Seoul did not isolate such a component. The NOA factor exhibited the highest nitrogen-to-carbon (N:C) ratio (0.22) and the lowest oxygen-to-carbon (O:C) ratio (0.19) among all

POA factors (Fig. S2), indicating a chemically reduced, nitrogen-rich composition. The NOA mass spectrum was dominated by amine-related fragments including $m/z$ 30 ($CH_4N^+$), 44 ($C_2H_6N^+$), 58 ($C_3H_8N^+$), and 86 ($C_5H_{12}N^+$) (Fig. 3a). The spectral signature of the factor is defined by the characteristic dominance of the $m/z$ 44 fragment, which typically serves as the primary marker for dimethylamine (DMA)-related species, closely followed by $m/z$ 58 (trimethylamine, TMA) and $m/z$ 30 (methylamine, MA). This profile is in strong agreement with NOA factors resolved via PMF in other polluted environments. For instance, the dominance of $m/z$ 44 and $m/z$ 30 aligns with amine factors reported in New York City (Sun et al., 2011) and Pasadena, California (Hayes et al., 2013). This DMA-dominated signature is also consistent with seasonal characterization of organic nitrogen in Beijing (Xu et al., 2017) and Po Valley, Italy (Saarikoski et al., 2012), reinforcing the common chemical signature of reduced organic nitrogen across diverse urban and regional environments.

In this study, NOA contributed approximately 2 % of total OA, comparable to urban contributions reported in Guangzhou (3 %; Chen et al., 2021), Pasadena (5 %; Hayes et al., 2013), and New York (5.8 %; Sun et al., 2011). These similarities suggest that the NOA factor observed in Seoul reflects a broader class of urban wintertime reduced-nitrogen aerosols rather than a site-specific anomaly. Furthermore, the presence of non-negligible signals at m/z 58 and m/z 86 supports the contribution of slightly larger alkylamines, a pattern that aligns well with established AMS laboratory reference spectra (Ge et al., 2011; Silva et al., 2008). In most urban environments, the detectability of NOA appears to depend strongly on the interplay between emission strength, stagnation, and humidity—which together govern the particle-phase partitioning of volatile amines.

Line 442: The decoupling between O:C ratio and volatility has been reported previously in the literature. Consider moderating the wording by removing "striking" and instead emphasizing that this behavior is newly observed or documented for Seoul

We appreciate this constructive comment. We have moderated the language by replacing "striking" with **"notable"** and have emphasized that this is a newly documented observation for the Seoul metropolitan area. The revised sentence reads:

Concurrently, the volatility analysis revealed a notable decoupling between oxidation state and volatility for the More-Oxidized Oxygenated OA (MO-OOA).